# Supervised Domain Adaptation Based on Marginal and Conditional Distributions Alignment

**Ori Katz**                                                                              *orikats@gmail.com*
*Viterbi Faculty of Electrical and Computer Engineering*
*Technion – Israel Institute of Technology*

**Ronen Talmon**                                                                  *ronen@ee.technion.ac.il*
*Viterbi Faculty of Electrical and Computer Engineering*
*Technion – Israel Institute of Technology*

**Uri Shaham**                                                                      *uri.shaham@biu.ac.il*
*Department of Computer Science*
*Bar-Ilan University*

**Reviewed on OpenReview:** *https://openreview.net/forum?id=ffBj12yh58*

## Abstract

Supervised domain adaptation (SDA) is an area of machine learning, where the goal is to achieve good generalization performance on data from a target domain, given a small corpus of labeled training data from the target domain and a large corpus of labeled data from a related source domain. In this work, based on a generalization of a well-known theoretical result of Ben-David et al. (2010), we propose an SDA approach, in which the adaptation is performed by aligning the marginal and conditional components of the input-label joint distributions. In addition to being theoretically grounded, we demonstrate that the proposed approach has two advantages over existing SDA approaches. First, it applies to a broad collection of learning tasks, such as regression, classification, multi-label classification, and few-shot learning. Second, it takes into account the geometric structure of the input and label spaces. Experimentally, despite its generality, our approach demonstrates on-par or superior results compared with recent state-of-the-art task-specific methods. Our code is available here.

## 1 Introduction

The Empirical Risk Minimization (ERM) principle, which is the theoretical basis of supervised machine learning, is based on the assumption that the training data is sampled from the data distribution that the model will encounter after deployment. Unfortunately, in practice, this assumption does not hold in many applications and tasks, and often significant discrepancies between the train and test distributions are prevalent. Mitigating these discrepancies is systematically addressed by domain adaptation (DA). In the literature, many contexts of DA are considered. Here, we focus on supervised domain adaptation (SDA) in the following setting. Given a large corpus of labeled data from a source domain and a small corpus of labeled data from a target domain, the goal is to facilitate learning with a good generalization on data from the target domain. This setting is highly suitable for scenarios such as emerging domains (e.g. early stages of a pandemic), rare events (e.g. natural disasters), costly data acquisition (e.g. remote sensing or certain types of environmental monitoring), and privacy concerns (e.g. in healthcare or finance sectors, where data privacy regulations may limit even unlabeled data use). These challenges necessitate effective SDA approaches.

In their seminal work, Ben-David et al. (2010) presented an upper bound on the generalization error on the target domain for binary classification and deterministic labeling functions. Here, we generalize this result (i) by considering arbitrary supervised learning tasks, and (ii) by defining the domain as an input-label

joint distribution. The derived upper bound consists of two terms. The first term, often referred to as the covariate shift, conveys the discrepancy between the marginal distributions on the input space. The second term represents the discrepancy between the conditional distributions on the label space, and therefore, we term it as the cross-domain conditional agreement (CDCA) error. The covariate shift has been widely investigated in the literature of *unsupervised* DA (UDA) Pan et al. (2010); Long et al. (2018); Sun et al. (2017); Zhang et al. (2019b), and many of the existing algorithms for UDA attempt to minimize it. These methods aim to mitigate this shift by mapping the input space into a representation space of invariant features across domains. However, ensuring similarity in the conditional distribution of the label space given the mapped input space remains challenging Gong et al. (2016). To face this gap, recent research in DA has also explored aligning conditional distributions. Gong et al. (2016) proposed an unsupervised method that focuses on aligning conditional transferable components, considering a causal system in which labels are the cause for inputs. Under the assumption that the conditional distribution of the transferable components is invariant after a location-scale transformation, Gong et al. (2016) devised a UDA method for classification tasks. A similar approach, addressing the alignment of the conditional distributions, was proposed by Richard et al. (2021). Inspired by the bound proposed in Ben-David et al. (2010), they derived a bound based on the conditional distribution of the input space given the considered hypothesis function and devised a method for finding a hypothesis function minimizing the derived bound. Considering the conditional distributions has been utilized in multi-source domain adaptation as well. For example, Heinze-Deml & Meinshausen (2021) addressed the challenges raised by domain shifts in image classification and introduced conditional variance penalties to enhance domain-shift robustness. Although this example is not directly related to conditional alignment, it underscores the broader relevance of considering conditional distributions into account in the DA landscape.

In contrast to UDA, existing methods for SDA are not driven by the theoretical bound of Ben-David et al. (2010), and both the covariate shift and the CDCA term have been overlooked. More specifically, the main paradigm of SDA methods, which are designed specifically for classification tasks, is to employ heuristics that aim to map samples from the same class to a single point, ignoring the distribution of the samples in the input space.

While our generalization of the upper bound has its own merit, we show that it introduces a new SDA approach, in which the input space and the label space are decoupled. Specifically, based on the covariate shift and the CDCA error terms, we formulate the problem of SDA as a multi-term optimization problem and propose to solve it using a Siamese neural network. In our experiments, we demonstrate the broad applicability of the proposed method to classification, regression, multi-label classification, and few-shot learning tasks, where we show on-par or superior results compared to recent state-of-the-art methods that are specific for each task.

The main contributions of this paper are as follows. First, we present a new approach to SDA that is theoretically grounded and widely applicable. Second, we extend the upper bound of the generalization error in the target domain to broader and more practical scenarios. Third, driven by the extended upper bound, we introduce a new loss term, and subsequently, a geometry-aware SDA method that is based on a Siamese neural network.

## 2 Related Work

The task of SDA lies in the interface of UDA, multi-task learning (MTL) Zhang et al. (2019a); Dakota et al. (2021), and multi-source domain adaptation (MSDA) Sun et al. (2015), and it is highly related to continual learning Tang et al. (2021) and catastrophic forgetting Thompson et al. (2019); Li et al. (2022).

While UDA focuses on transferring knowledge solely between the input spaces (marginal distributions of inputs), in MTL the ultimate goal is to improve performance on multiple related tasks by sharing information essential to these tasks. Therefore, MTL focuses on the distributions of the label space, often assuming the same input space distribution. SDA leverages both the input space and the label space (i.e., the joint distributions of inputs and labels), allowing exploitation of the relatedness between source and target domains, even when the joint distributions slightly differ. Another highly related field is MSDA Sun et al. (2015). While MSDA can be applied to SDA problems by treating each source domain as a separate task,

SDA typically focuses on scenarios with a single source domain and a limited amount of labeled data in the target domain. This focus on few-shot learning from the target domain typically differentiates SDA from MSDA, often necessitating more complex machinery to handle data from multiple sources.

SDA is also related to continual learning Tang et al. (2021) and catastrophic forgetting Thompson et al. (2019); Li et al. (2022). In continual learning, the goal is to train a model on a sequence of tasks while preserving the knowledge acquired from previous tasks. This is similar to SDA in the sense that both approaches aim to handle evolving data distributions. However, continual learning typically deals with a single domain and focuses on mitigating forgetting during training, whereas SDA addresses adaptation between two different domains.

Since our SDA approach shares principles with UDA approaches, we begin this section with a survey of UDA methods. Then, we focus on SDA techniques.

## 2.1 Unsupervised Domain Adaptation

The large majority of methods for UDA can be broadly divided into two paradigms – instance-based methods and feature-based methods. The *instance-based paradigm* is based on samples re-weighting. The core idea behind this paradigm is to approximate the target error using a weighted sum of source errors. One notable shortcoming of instance-based methods is that they assume a large overlap between the supports of the source and target densities, and hence, are limited when the covariate shift is relatively large Teshima et al. (2020). In these cases, the second paradigm of *feature-based methods* becomes preferable. The common trait shared by methods in the feature-based paradigm is the application of a mapping function to the input features. Early methods in UDA attempt to minimize the covariate shift by finding a mapping from the target distribution to the source distribution Gopalan et al. (2011); Gong et al. (2012); Fernando et al. (2013). A different, yet related approach, which is often used for domain generalization, is to find a latent space in which the representation is domain-invariant. Long et al. (2013); Pan et al. (2010). When the samples from the source domain are labeled, the labels are also taken into account. In the last decade, due to the emergence of deep neural networks, utilizing Siamese neural networks (NN) for domain adaptation has become the leading technique to incorporate the source labels, e.g. Tzeng et al. (2014); Sun et al. (2017); Ganin & Lempitsky (2015); Tzeng et al. (2017; 2015); Long et al. (2018; 2015; 2017). In some methods, domain invariance is obtained by incorporating probability measures such as maximum mean discrepancy (MMD) Tzeng et al. (2014), multi-kernel MMD Long et al. (2015), joint MMD Long et al. (2017), or even second-order statistics Sun et al. (2017). Recently, UDA has seen significant progress by incorporating methods utilizing adversarial approaches, e.g., Ganin et al. (2016); Tzeng et al. (2017); Long et al. (2018); Zhang et al. (2019b). These methods leverage generative adversarial networks, where a generator aiming to extract domain-agnostic features is pitted against a discriminator aiming to distinguish between features extracted from the source domain and features extracted from the target domain. This adversarial training is carried out by a minimax optimization algorithm and encourages the generator to learn a shared feature space where the domains become indistinguishable. This idea was first introduced in Ganin et al. (2016), generalized in Tzeng et al. (2017), and paved the way for many techniques. For example, Long et al. (2018) proposed conditioning the generator on class labels, promoting better adaptation for classification tasks. Particularly relevant to our work is Zhang et al. (2019b), who extended the theoretical result of Ben-David et al. (2010) to multi-class classification and introduced the Margin Disparity Discrepancy (MDD), a novel measurement with rigorous generalization bounds, measuring the discrepancy between the source and target domains in terms of their classification margins. The results of Zhang et al. (2019b) established a theoretical foundation for adversarial UDA methods, bridging the gap between Ben-David et al. (2010)'s theory and a practical UDA algorithm.

## 2.2 Supervised Domain Adaptation

While minimizing the covariate shift is a compromise when no labels from the target domain are at hand, one could argue that same approaches for UDA can be employed in SDA in order to match the conditional distributions as well. However, when facing a small amount of data points from the target domain, this is unfortunately not the case. Therefore, existing methods for SDA employ other strategies. These strategies

can be broadly divided according to the downstream task, or more specifically, according to the marginal distributions of labels. In recent years, a substantial body of literature in SDA addressed classification tasks, while regression tasks have gained less attention. For *regression*, the main strategy is re-weighting, e.g., Yao & Doretto (2010); Pardoe & Stone (2010). Recently, in accordance with many UDA methods, employing NNs combined with an adversarial mechanism was proposed for the weights selection de Mathelin et al. (2021). In Teshima et al. (2020), the authors proposed a new SDA approach for regression tasks that does not fall into the previously mentioned paradigms, and is based on data augmentation, where the target dataset is augmented using the learned statistics from the source dataset. In *classification*, the main strategy can be viewed as a feature-based approach. The main paradigm for SDA classification is contrastive-based, e.g., Motiian et al. (2017b;a). The objective of these methods is to map samples from the same class to a single point while separating the points representing different classes. Note that in contrast to UDA, these methods do not explicitly aim to minimize the covariate shift or the CDCA term. Moreover, their objective often contracts the within-class geometry and ignores the between-class geometry. The second paradigm for SDA classification is geometry-based, e.g., Xu et al. (2019); Morsing et al. (2021), in which a locally geometry-aware loss is employed. For example, in Xu et al. (2019) this is a modified-Hausdorff distance which is employed via stochastic neighborhood embedding, while in Morsing et al. (2021) this is a modification of the Rayleigh quotient. Note that although being geometry-aware, these methods do not aim to preserve between-class and within-class geometries.

## 3  Learning Setup and Problem Formulation

We adopt the formulation of Ben-David et al. (2010) with some modifications. Let $\mathcal{X}$ be an input space and $\mathcal{Y}$ be the label space. We will use the terms label space and output space interchangeably to emphasize that our work is not restricted to discrete or categorical label spaces.

**Definition 3.1** (Domain)**.** A joint distribution $\mathcal{P}$ defined on the input-output product space $\mathcal{X} \times \mathcal{Y}$ is called a *domain*. We assume that a domain $\mathcal{P}$ has a density function, denoted by $P(x, y)$.

**Definition 3.2** (Hypothesis and Error)**.** A *hypothesis* is a function $h : \mathcal{X} \to \mathcal{Y}$. The *error of a hypothesis* $h$ with respect to a domain $\mathcal{P}$ is $\epsilon_{\mathcal{P}}(h) \triangleq \mathbb{E}_{(x,y)\sim\mathcal{P}} \, \ell(h(x), y)$, where $\ell : \mathcal{Y} \times \mathcal{Y} \to \mathbb{R}$ is a loss function that is typically determined by the specific downstream task.

Consider a source domain $\mathcal{P}^S$ and a target domain $\mathcal{P}^T$, and consider $N^S$ samples $\{(x_i^S, y_i^S)\}_{i=1}^{N^S}$ drawn from the source domain $\mathcal{P}^S$ and $N^T$ samples $\{(x_i^T, y_i^T)\}_{i=1}^{N^T}$ drawn from the target domain $\mathcal{P}^T$, where $(x_i^v, y_i^v) \overset{\text{iid}}{\sim} \mathcal{P}^v$ for $v \in \{S, T\}$. Following the common practice, we assume that $N^T \ll N^S$. Our goal is to find a hypothesis $h^*$, such that: $h^* \triangleq \operatorname{argmin}_h \epsilon_{\mathcal{P}^T}(h)$.

## 4  Bounding the Error of the Target Domain

Ben-David et al. (2010) presented an upper bound of the target error of a hypothesis. Their bound is derived for deterministic labeling functions and binary output spaces. We generalize their result to (i) the definition of a domain as a joint distribution and (ii) arbitrary supervised learning tasks.

Let $\mathcal{Q}^v$ denote the marginal distribution over the input space in domain $v \in \{S, T\}$ equipped with the density function $Q^v(x) = \int_{y \in \mathcal{Y}} P^v(x, y) dy$ for any $x \in \mathcal{X}$. Let $d(\mathcal{Q}^S, \mathcal{Q}^T)$ denote the total variation distance between the marginals, given by: $d(\mathcal{Q}^S, \mathcal{Q}^T) \triangleq \sup_{B \in \mathcal{B}} |Q^S(B) - Q^T(B)|$, where we define, with a slight abuse of notation, $Q^v(B) = \int_{x \in B} Q^v(x) dx$ and $\mathcal{B}$ is the $\sigma$-algebra of all measurable subsets in the support of $Q^S$ and $Q^T$. We define the cross-domain conditionals agreement (CDCA) error of a domain $v \in \{S, T\}$ by:

$$\chi^v \triangleq \underset{x \sim \mathcal{Q}^v}{\mathbb{E}} \underset{\substack{y \sim \mathcal{P}_x^S \\ y' \sim \mathcal{P}_x^T}}{\mathbb{E}} \ell(y, y') \tag{1}$$

where $\mathcal{P}_x^v$ denotes the conditional distribution of the output space given $x \in \mathcal{X}$ in domain $v \in \{S, T\}$.

We further assume that there exists an upper bound, denoted by $M$, for the expectation of the loss $M \triangleq \max\{\sup_h \epsilon_{\mathcal{P}^S}(h), \sup_h \epsilon_{\mathcal{P}^T}(h)\}$. Note that, for example, if $\ell$ is the $0 - 1$ loss, then $M \leq 1$. We remark that

for the Cross-entropy (CE) loss this assumption does not hold, however, since CE is used as a surrogate for the intractable $0 - 1$ loss, it does not affects the generality of this assumption.

**Theorem 4.1.** *For any hypothesis function $h : \mathcal{X} \to \mathcal{Y}$ and loss function satisfying: $|\ell(x, y) - \ell(x, z)| \leq |\ell(y, z)|$, we have:*

$$\epsilon_{\mathcal{P}^T}(h) \leq \epsilon_{\mathcal{P}^S}(h) + Md(\mathcal{Q}^T, \mathcal{Q}^S) + \min\{\chi^S, \chi^T\}. \tag{2}$$

The proof is in Appendix A. Note that the requirement on the loss function in Theorem 4.1 is satisfied by frequently-used loss functions, such as the $0 - 1$, L1, and L2 losses. CE does not generally satisfy this assumption. However, in certain cases involving densities that result from CE minimization, it does Shore & Johnson (1981).

The derived upper bound in equation 2 consists of three terms. The first term, $\epsilon_{\mathcal{P}^S}(h)$, is the error of the hypothesis with respect to the source domain. The second term, $d(\mathcal{Q}^T, \mathcal{Q}^S)$, is originated from the covariate shift between the marginal distributions of the input space. The last term, $\min\{\chi^S, \chi^T\}$, conveys the discrepancy between the conditional distributions of the two domains.

## 5 Proposed Approach

### 5.1 From the Theory to an Optimization Algorithm

At first glance, the bound in Theorem 4.1 may be viewed only as a theoretical guarantee for the adaptation capability between two domains. However, a closer look from an algorithmic viewpoint suggests a practical utility. Specifically, consider a (measurable) mapping function $\phi$ from the input space $\mathcal{X}$ to some embedding space $\mathcal{E}$. Let $\widetilde{\mathcal{P}}^v$, $v \in \{S, T\}$, be the joint distribution defined on the embedded-output space that is induced by the marginal pushforward distribution, i.e., the joint distribution of $(\phi(x), y)$, where $(x, y) \sim \mathcal{P}^v$.

Considering the embedded space $\mathcal{E}$ as an analog of the input space $\mathcal{X}$ introduces another degree of freedom that depends on the mapping function $\phi$. Specifically, we can now use the upper bound in Theorem 4.1 to find a hypothesis *and* a mapping function $\phi$ that minimizes the following objective:

$$(\phi^*, \tilde{h}^*) = \underset{(\phi, \tilde{h})}{\operatorname{argmin}} \, \epsilon_{\widetilde{\mathcal{P}}^S}(\tilde{h}) + d(\widetilde{\mathcal{Q}}^S, \widetilde{\mathcal{Q}}^T) + \min\{\widetilde{\chi}^S, \widetilde{\chi}^T\}, \tag{3}$$

where $\widetilde{\mathcal{Q}}^v$ and $\widetilde{\chi}^v$ denote the marginal pushforward distribution and the corresponding CDCA error for $v \in \{S, T\}$, respectively, and $\tilde{h} : \mathcal{E} \to \mathcal{Y}$ is a hypothesis function defined on the embedded space.

Putting the optimization in equation 3 into practical use in the context of the considered SDA problem involves two elements. The first element is the approximation of the terms based on the available data at hand. A straightforward approximation of the error $\epsilon_{\widetilde{\mathcal{P}}^S}(\tilde{h})$ is given by: $\hat{\epsilon}_{\widetilde{\mathcal{P}}^S}(\tilde{h}) \triangleq \frac{1}{N_s} \sum_{i=1}^{N_s} \ell(\tilde{h}(\phi(x_i^S)), y_i^S)$. The approximation of the covariate shift, denoted by $\hat{d}(\widetilde{\mathcal{Q}}^S, \widetilde{\mathcal{Q}}^T)$, can be obtained by employing any UDA technique to the embedded training data via $\phi$. For example, Long et al. (2015) approximated this term using the MMD distance between $\{\phi(x_i^S)\}_{i=1}^{N^S}$ and $\{\phi(x_i^T)\}_{i=1}^{N^T}$.

The approximation of the term $\min\{\widetilde{\chi}^S, \widetilde{\chi}^T\}$ is more challenging. By observing the definition of the CDCA error from Equation (1), we see that the approximation of $\widetilde{\chi}^S$ requires input-label samples $(x, y)$ from the source domain, which are available, but also input-label samples $(x, y')$, where $x$ is sampled from the source domain $x \sim \mathcal{Q}^S$ but $y'$ from the marginal distribution induced by the target domain $\widetilde{\mathcal{P}}^T$. An analogous challenge is raised in the approximation of $\widetilde{\chi}^T$. We propose to address these challenges using kernel regression in Section 5.2.

The second practical element is the incorporation of the available labeled data from the target domain. So far, our focus was on the minimization of equation 3, which is based on the upper bound of the hypothesis error in the target domain in the r.h.s. of equation 2. Importantly, in our SDA setting, a small number of input-output samples from the target domain are available. Although they are not sufficient for accurate or meaningful learning, the available target samples facilitate an approximation of the error: $\hat{\epsilon}_{\widetilde{\mathcal{P}}^T}(\tilde{h}) \triangleq$

$\frac{1}{N_t} \sum_{i=1}^{N_t} \ell(\tilde{h}(\phi(x_i^T)), y_i^T)$. For any $0 \le \alpha \le 1$, we can express the bound from eq. (2) as follows:

$$\epsilon_{\widetilde{\mathcal{P}}^T}(h) = (1-\alpha)\epsilon_{\widetilde{\mathcal{P}}^T}(h) + \alpha\epsilon_{\widetilde{\mathcal{P}}^T}(h) \le (1-\alpha)\epsilon_{\widetilde{\mathcal{P}}^T}(h) + \alpha\big(\epsilon_{\widetilde{\mathcal{P}}^S}(h) + Md(\widetilde{\mathcal{Q}}^T, \widetilde{\mathcal{Q}}^S) + \min\{\widetilde{\chi}^S, \widetilde{\chi}^T\}\big).$$

which gives rise to the following optimization problem:

$$(\phi^*, \tilde{h}^*) = \underset{(\phi, \tilde{h})}{\operatorname{argmin}}(1-\alpha)\hat{\epsilon}_{\widetilde{\mathcal{P}}^T}(\tilde{h}) + \alpha(\hat{\epsilon}_{\widetilde{\mathcal{P}}^S}(\tilde{h}) + \operatorname{align}(\widetilde{\mathcal{P}}^S, \widetilde{\mathcal{P}}^T)), \qquad (4)$$

where: $\operatorname{align}(\widetilde{\mathcal{P}}^S, \widetilde{\mathcal{P}}^T) \triangleq \hat{d}(\widetilde{\mathcal{Q}}^S, \widetilde{\mathcal{Q}}^T) + \min\{\widetilde{\chi}^S, \widetilde{\chi}^T\}$. In practice, the selection of the hyper-parameter $\alpha$ reflects the prior belief for the capability to estimate $\tilde{h}$ from the available target samples. We remark that the proposed objective assumes the form of typical objective functions for SDA consisting of a source term, a target term, and an alignment term.

## 5.2 Implementation

We adopt a popular approach employed in DA and utilize a Siamese network architecture Motiian et al. (2017b); Xu et al. (2019). We consider the last layer as the hypothesis function $\tilde{h}$, and the rest of the network is viewed as the mapping function $\phi$ of the input data to an embedded space. During training, this network is simultaneously fed with pairs of mini-batches from the source and target domains.

Let $(\boldsymbol{X}^S, \boldsymbol{Y}^S), (\boldsymbol{X}^T, \boldsymbol{Y}^T)$ denote mini-batches of $n$ samples from the source and target training sets, respectively. We denote their mappings by $\phi(\boldsymbol{X}^v) \in \mathbb{R}^{n \times d}$ for $v = \{S, T\}$, where $d$ is a hyperparameter denoting the dimensionality of the output layer of $\phi$. In this setting, the implementation of $\hat{\epsilon}_{\widetilde{\mathcal{P}}^S}(\tilde{h})$ and $\hat{\epsilon}_{\widetilde{\mathcal{P}}^T}(\tilde{h})$ translates to the following loss term: $L(\tilde{h}(\phi(\boldsymbol{X}^T)), \boldsymbol{Y}^T) + L(\tilde{h}(\phi(\boldsymbol{X}^S)), \boldsymbol{Y}^S)$, where $L(\boldsymbol{X}, \boldsymbol{Y})$ is an aggregation operator of the considered loss function $\ell$: $L(\boldsymbol{X}, \boldsymbol{Y}) \triangleq \sum_{i=1}^{n} \ell(\boldsymbol{X}_i, \boldsymbol{Y}_i)$, and $\boldsymbol{X}_i$ and $\boldsymbol{Y}_i$ denotes the $i$th sample in $\boldsymbol{X}$ and $\boldsymbol{Y}$, respectively. The implementation of the empirical covariate shift, $\hat{d}(\widetilde{\mathcal{Q}}^S, \widetilde{\mathcal{Q}}^T)$, is given by $L_{\text{UDA}}(\phi(\boldsymbol{X}^S), \phi(\boldsymbol{X}^T))$, where $L_{\text{UDA}}$ denotes the loss term of the chosen UDA technique.

In our experiments, we used the UDA method proposed in Sun et al. (2017), which attempts to minimize the covariate shift by aligning the second-order statistics of the source and target distributions. This method does not require hyperparameters tuning, is easy to apply, and according to a recent survey achieves state-of-the art performance on standard benchmarks (see Table I in Preciado-Grijalva & Muthireddy (2021)). We note that we also tested other UDA techniques, e.g., MMD Tzeng et al. (2014) and the adversarial method from Tzeng et al. (2017), which led to similar results.

The CDCA term in equation 4 is approximated by using kernel regression (KR) Watson (1964), aiming to generate labels from a certain domain given the mapped features and labels of its cross-domain. More specifically, let $\bar{v}$ denote the cross-domain of $v \in \{S, T\}$, we use KR in order to generate $\hat{y}$ that admits the distribution of $\widetilde{\mathcal{P}}_x^{\bar{v}}$ where $x \sim \mathcal{Q}^v$. Then we use these generated samples in order to generate pairs of $(\phi(x), \hat{y})$ to approximate $\widetilde{\chi}^v$. The exact procedure is described in Algorithm 1. We remark that the motivation for using kernel regression, rather than a standard parametric estimator like a neural network, is to avoid overfitting due to the small number of available training samples from the target domain. This choice raises computational complexity and limitations, which we further discuss in Appendix D.

Let $\widehat{\boldsymbol{Y}}^v, v \in \{S, T\}$, denote the output of the application of Algorithm 1 to $\phi(\boldsymbol{X}^v)$, $\phi(\boldsymbol{X}^{\bar{v}})$, and $\boldsymbol{Y}^{\bar{v}}$. Then, the computation of the CDCA term is given by: $\min\{L(\boldsymbol{Y}^S, \widehat{\boldsymbol{Y}}^S), L(\boldsymbol{Y}^T, \widehat{\boldsymbol{Y}}^T)\}$.

To conclude, the proposed method includes a Siamese NN, as presented in Figure 1. During training, the network is fed with pairs of labeled mini-batches: $(\boldsymbol{X}^S, \boldsymbol{Y}^S)$ and $(\boldsymbol{X}^T, \boldsymbol{Y}^T)$. The training objective, analogous to equation 4, is to minimize the following loss term:

$$L(\tilde{h}(\phi(\boldsymbol{X}^T)), \boldsymbol{Y}^T) + L(\tilde{h}(\phi(\boldsymbol{X}^S)), \boldsymbol{Y}^S) + L_{\text{align}}(\phi(\boldsymbol{X}^S), \boldsymbol{Y}^S, \phi(\boldsymbol{X}^T), \boldsymbol{Y}^T), \qquad (5)$$

where the alignment term $L_{\text{align}}$, analogous to $\operatorname{align}(\widetilde{\mathcal{P}}^S, \widetilde{\mathcal{P}}^T)$ in equation 4, is given by:

$$L_{\text{align}}(\phi(\boldsymbol{X}^S), \boldsymbol{Y}^S, \phi(\boldsymbol{X}^T), \boldsymbol{Y}^T) = L_{\text{UDA}}(\phi(\boldsymbol{X}^S), \phi(\boldsymbol{X}^T)) + \min\{L(\boldsymbol{Y}^S, \widehat{\boldsymbol{Y}}^S), L(\boldsymbol{Y}^T, \widehat{\boldsymbol{Y}}^T)\}.$$

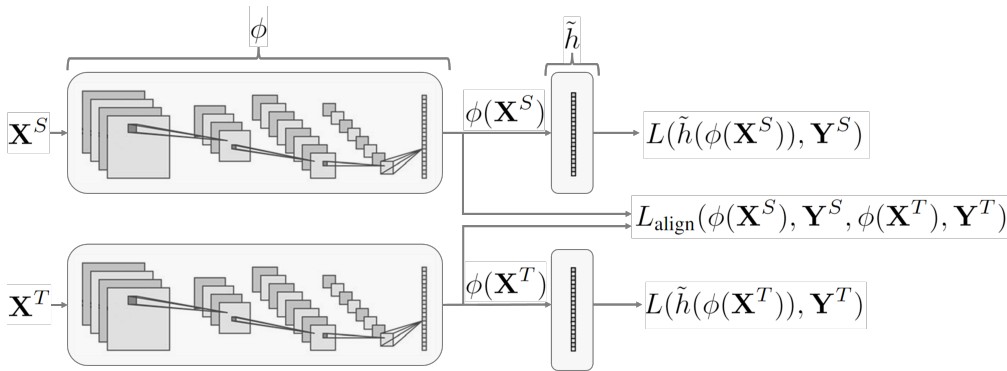

Figure 1: A schematic illustration of our proposed approach.

In practice, the terms in the proposed loss can be weighted according to hyperparameters tuning.

---

**Algorithm 1** Cross-domain samples generation via KR

---

**Input**: 1. Two mapped features of mini-batches, one from each domain $\phi(\boldsymbol{X}^v) \in \mathbb{R}^{d \times n}, v \in \{S, T\}$. 2. The labels for the mini-batch of the cross-domain $\boldsymbol{Y}^{\bar{v}}$.

**Output**: A mini-batch sample $\widehat{\boldsymbol{Y}}^v$, where each $y \in \widehat{\boldsymbol{Y}}^v$ admits the distribution of $\widetilde{\mathcal{P}}_x^{\bar{v}}$, where $x \sim \mathcal{Q}^v$.

1: Compute $\boldsymbol{D}_{v,\bar{v}}[i,j] = \|\phi(X_i^v) - \phi(X_j^{\bar{v}})\|_2$, for $i, j = 1, \ldots, n$, where $X_i^v$ denotes the $i$th row of $\boldsymbol{X}^v$ (corresponding to the $i$th sample in the batch).

2: Compute: $\boldsymbol{K}_{v,\bar{v}}[i,j] = \exp\left(-\frac{\boldsymbol{D}_{v,\bar{v}}^2[i,j]}{\epsilon_v^2}\right)$, where $\epsilon_v = \max_i \min_j \boldsymbol{D}_{v,\bar{v}}[i,j]$.

3: Approximate $\widehat{\boldsymbol{Y}}^v$ using kernel-based weighted average: $\widehat{\boldsymbol{Y}}^v = (\text{diag}(\boldsymbol{K}_{v,\bar{v}}\mathbf{1}))^{-1} \boldsymbol{K}_{v,\bar{v}} \boldsymbol{Y}^{\bar{v}}$, where $\mathbf{1}$ is a column vector of all ones.

---

# 6 Experimental Study

Our experimental study begins in Section 6.1 with commonly-used benchmarks for classification. We compare our approach to state-of-the-art (SOTA) methods for classification and show that our approach obtains on-par or superior results. In Section 6.2, we focus on scenarios that highlight the benefits stemming from taking the geometry of the input and output spaces into account. We present new datasets and learning tasks and demonstrate superior results compared to the SOTA baseline methods used in Section 6.1. We complete this section with an evaluation of regression tasks in Section 6.3. There, we show superior results compared to SOTA methods for regression, demonstrating the universality of our method and its broad applicability. In summary, throughout this section, we demonstrate on-par or superior results on multiple tasks compared to the SOTA methods tailored specifically for the respective task (i.e., classification or regression).

## 6.1 Classification on benchmark tasks

We consider three commonly-used benchmark tasks for DA. The "Digits" task consists of two domains (datasets): the MNIST dataset LeCun et al. (2010), denoted by $\mathcal{M}$, and the USPS dataset LeCun et al. (1989), denoted by $\mathcal{U}$. The "Office" task Saenko et al. (2010) consists of 3 domains: Amazon, Webcam, and DSLR, denoted by $\mathcal{A}, \mathcal{W}$, and $\mathcal{D}$, respectively. The "VisDA-C" task Peng et al. (2018) consists of two domains: a synthetic domain of 3D rendered objects, denoted by $\mathcal{S}$, and a real domain of images-in-the-wild, denoted by $\mathcal{R}$. An illustration of the datasets is presented in Figure 2. We compare our method to three state-of-the-art SDA methods. The first method, termed CCSA Motiian et al. (2017b), represents contrastive-based approaches. The second method, d-SNE Xu et al. (2019), represents geometric approaches. The third method, termed NEM Wang et al. (2019) is a combination of these two approaches. All of these methods share the same objective of mapping samples from the same class to a single point while separating the points representing different classes. In addition to these methods, we compare our approach with its

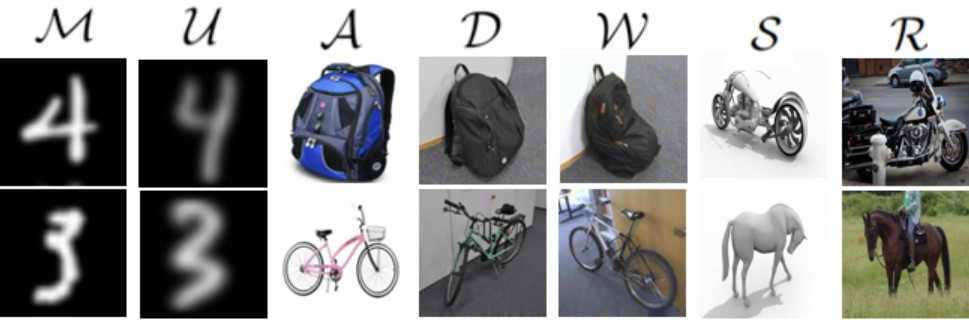

Figure 2: Illustrative samples from the "Digits" ($\mathcal{M}$ and $\mathcal{U}$), "Office" ($\mathcal{A}$, $\mathcal{W}$ and $\mathcal{D}$) and the "VisDA-C" ($\mathcal{S}$ and $\mathcal{R}$) tasks.

ablated versions. In the first ablation, denoted by "Ab-UDA", we discard the UDA loss term $L_{\mathrm{UDA}}$, and in the second ablation, denoted by "Ab-CDCA", we discard the CDCA term. For reference, we further report the results of three naïve baselines that include training using the source or target domain only (denoted by S-Only or T-Only, respectively) and simultaneously training on both domains (denoted by S+T).

In the first experiment, we consider the "Digits" tasks. We follow the same experimental protocol from Motiian et al. (2017b); Xu et al. (2019) (see details in Appendix C). We conduct several experiments, and in each experiment, we consider a different number of samples per class from the target domain. We repeat each experiment 10 times. In this experiment, 200 samples per class from the source domain and $X$ samples per class from the target domain are randomly selected as the training set, where $X \in \{1, 3, 5, 7\}$. We note that in most real-world problems, the data from the source domain is considered to be unlimited. Still, for the sake of a fair comparison, we chose to adopt the common evaluation protocol employed by the considered baselines Motiian et al. (2017b); Xu et al. (2019). In the sequel, we conduct experiments where we consider the entire source dataset for evaluation and observe similar results. The average results and their standard deviations are presented in Table 1. We note that we reproduced the baselines' results, see Appendix B for details. Bold indicates the best results, and underline indicates the second-best. In the table, we see that our proposed approach achieves on-par results with the compared baselines. In addition, we see the contribution of each term in the proposed loss in equation 5. Specifically, we see that the contribution of the UDA and the CDCA terms is especially pronounced when considering fewer samples per class.

In the second experiment, we consider the "Office" tasks. We consider six DA tasks: $\mathcal{A} \to \mathcal{D}, \mathcal{A} \to \mathcal{W}, \mathcal{D} \to \mathcal{A}, \mathcal{D} \to \mathcal{W}, \mathcal{W} \to \mathcal{A}$ and $\mathcal{W} \to \mathcal{D}$. We follow the same experimental protocol from Motiian et al. (2017b); Xu et al. (2019). For the source domain, we randomly pick 20 samples per class from Amazon and 8 samples per class from Webcam and DSLR. For the target domain, we randomly pick 3 samples per class. For more details, see Appendix C. We repeat this procedure 5 times, the average results and their standard deviations are presented in Table 2. Same as in Table 1, we see that our approach attains on-par results with the current SOTA methods.

In the third experiment, we consider a DA task from the synthetic domain ($\mathcal{S}$) of the 3D rendered objects in "VisDA-C" to the real domain ($\mathcal{R}$). We follow the experimental protocol from Xu et al. (2019). For more details, see Appendix C. We conduct several experiments, and in each experiment, we consider a different number of samples per class from the target domain. We repeat each experiment 10 times, the average results and their standard deviations are presented in Table 3. Here as well we see that our approach achieves on-par results with the current SOTA methods.

## 6.2 Leveraging the intrinsic geometry of a dataset

In this section, we highlight the scenarios in which our method is expected to excel and validate it empirically. In the first experiment, we present a variation of the SDA task. We randomly pick some classes and discard their associated training samples from the target domain. For the remaining target classes, we take one sample per class. We term the target classes we discard "cold" target classes and this entire task zero-

Table 1: Classification accuracy for the "Digits" tasks. In the upper part, we consider adaptation from $\mathcal{U}$ to $\mathcal{M}$. In the lower part, we consider the opposite direction. In each column, we test a different number of samples per class from the target domain.

|  |  | 1 | 3 | 5 | 7 | #Best |
|---|---|---|---|---|---|---|
| $\mathcal{U} \to \mathcal{M}$ | S-Only | $59.35 \pm 2.55$ | $63.74 \pm 1.76$ | $59.56 \pm 1.59$ | $60.02 \pm 2.57$ | 0 |
|  | T-Only | $46.26 \pm 0.98$ | $65.24 \pm 2.13$ | $76.85 \pm 1.79$ | $82.73 \pm 1.26$ | 0 |
|  | S+T | $77.40 \pm 1.31$ | $85.99 \pm 1.44$ | $87.35 \pm 1.36$ | $\underline{91.68 \pm 0.78}$ | 0 |
|  | CCSA | $\mathbf{82.40 \pm 1.56}$ | $\mathbf{88.00 \pm 1.24}$ | $\underline{90.05 \pm 1.02}$ | $90.82 \pm 1.21$ | 2 |
|  | NEM | $75.75 \pm 1.82$ | $83.76 \pm 1.43$ | $88.27 \pm 1.29$ | $89.70 \pm 1.14$ | 0 |
|  | d-SNE | $79.42 \pm 1.88$ | $85.35 \pm 1.16$ | $89.09 \pm 0.93$ | $90.93 \pm 0.91$ | 0 |
|  | Ours | $\underline{81.67 \pm 1.73}$ | $\underline{86.31 \pm 1.07}$ | $\mathbf{90.20 \pm 1.14}$ | $\mathbf{92.08 \pm 0.50}$ | 2 |
|  | Ab-UDA | $74.24 \pm 1.67$ | $84.43 \pm 1.77$ | $87.78 \pm 1.51$ | $91.09 \pm 0.71$ | 0 |
|  | Ab-CDCA | $78.43 \pm 1.80$ | $85.60 \pm 1.38$ | $89.38 \pm 0.91$ | $90.84 \pm 1.04$ | 0 |
| $\mathcal{M} \to \mathcal{U}$ | S-Only | $78.07 \pm 1.04$ | $78.04 \pm 1.25$ | $78.25 \pm 1.13$ | $76.72 \pm 1.47$ | 0 |
|  | T-Only | $61.87 \pm 1.58$ | $76.46 \pm 1.63$ | $81.18 \pm 1.41$ | $84.41 \pm 1.28$ | 0 |
|  | S+T | $82.41 \pm 1.50$ | $87.94 \pm 1.11$ | $89.23 \pm 1.38$ | $90.14 \pm 0.77$ | 0 |
|  | CCSA | $\underline{85.11 \pm 1.23}$ | $87.51 \pm 1.15$ | $89.60 \pm 0.94$ | $90.35 \pm 0.71$ | 0 |
|  | NEM | $84.38 \pm 1.22$ | $87.16 \pm 1.32$ | $89.88 \pm 1.01$ | $90.54 \pm 1.15$ | 0 |
|  | d-SNE | $84.39 \pm 1.42$ | $\underline{89.04 \pm 1.14}$ | $89.42 \pm 1.05$ | $91.34 \pm 0.88$ | 0 |
|  | Ours | $\mathbf{85.85 \pm 0.87}$ | $\mathbf{89.46 \pm 0.90}$ | $\mathbf{90.80 \pm 0.67}$ | $\mathbf{91.52 \pm 0.88}$ | 4 |
|  | Ab-UDA | $83.09 \pm 1.48$ | $87.67 \pm 1.34$ | $89.46 \pm 0.69$ | $91.05 \pm 0.83$ | 0 |
|  | Ab-CDCA | $84.51 \pm 1.18$ | $87.35 \pm 1.03$ | $\underline{90.62 \pm 0.90}$ | $\underline{91.46 \pm 0.80}$ | 0 |

Table 2: Classification accuracy for the "Office" tasks. In each column, we consider a different task.

|  | $\mathcal{A} \to \mathcal{D}$ | $\mathcal{A} \to \mathcal{W}$ | $\mathcal{D} \to \mathcal{A}$ | $\mathcal{D} \to \mathcal{W}$ | $\mathcal{W} \to \mathcal{A}$ | $\mathcal{W} \to \mathcal{D}$ | #Best |
|---|---|---|---|---|---|---|---|
| S-Only | $57.86 \pm 1.75$ | $54.11 \pm 1.62$ | $50.10 \pm 1.16$ | $90.05 \pm 1.36$ | $48.39 \pm 1.24$ | $92.77 \pm 1.66$ | 0 |
| T-Only | $82.03 \pm 1.00$ | $81.38 \pm 1.52$ | $59.14 \pm 1.46$ | $79.74 \pm 1.45$ | $59.26 \pm 1.53$ | $84.38 \pm 1.73$ | 0 |
| S+T | $82.81 \pm 1.35$ | $\underline{84.74 \pm 1.14}$ | $\mathbf{66.44 \pm 0.85}$ | $93.02 \pm 0.61$ | $64.51 \pm 0.91$ | $\underline{97.27 \pm 1.00}$ | 1 |
| CCSA | $70.41 \pm 1.52$ | $71.72 \pm 1.27$ | $59.33 \pm 1.37$ | $\mathbf{94.37 \pm 0.86}$ | $57.26 \pm 1.07$ | $94.53 \pm 1.21$ | 1 |
| NEM | $\underline{84.77 \pm 1.61}$ | $\mathbf{85.02 \pm 1.02}$ | $63.44 \pm 1.12$ | $\underline{94.32 \pm 1.20}$ | $64.32 \pm 1.33$ | $96.96 \pm 0.94$ | 1 |
| d-SNE | $83.50 \pm 2.25$ | $84.58 \pm 1.52$ | $65.52 \pm 0.99$ | $94.06 \pm 0.77$ | $\underline{64.59 \pm 1.33}$ | $96.35 \pm 1.01$ | 0 |
| Ours | $\mathbf{87.56 \pm 1.65}$ | $83.50 \pm 0.82$ | $\underline{65.56 \pm 0.94}$ | $93.08 \pm 1.05$ | $\mathbf{66.28 \pm 1.40}$ | $\mathbf{97.42 \pm 0.75}$ | 3 |

shot SDA. We use the datasets from the "Digits" tasks as in Section 6.1 and follow the same experimental protocol. In this experiment, 200 samples per class from the source domain and one sample per class from the target domain are randomly selected as the training set. In addition, in each experiment, we consider a different number of "cold" target classes which are then ablated from the training set. We remark that in the remainder of this section, the entire source datasets are utilized for the evaluation. For brevity, we consider two representative baselines from Section 6.1: CCSA Motiian et al. (2017b), a contrastive-based approach, and d-SNE Xu et al. (2019), a geometry-based approach. The results are presented in Table 4. First, comparing d-SNE to CCSA, we see a clear benefit of incorporating geometric considerations into the model's design. Next, comparing our approach with d-SNE, we see that incorporating the marginals distribution alignment into the loss term attains even more significant improvements, especially when considering a large number of "cold" classes. This improvement is evident both in the average accuracy and in the standard deviations.

In the second experiment, we test a multi-label classification task in a zero-shot setting that is based on the "Digits" datasets. We consider an input space that consists of a concatenation of 7 digit slots, where each slot can either contain an instance of a digit or be blank. The associated output space consists of binary indicator vectors, where the $i$th entry indicates whether the $i$th digit appeared in the concatenated image. The concatenated images in the source domain consist of MNIST digits and the concatenated images in the target domain consist of USPS digits. In addition, the source domain admits a zero-shot setting, where the

Table 3: Classification accuracy for the "VisDA-C" task. In each row, we test a different number of samples per class from the target domain.

| | S-Only | T-Only | S+T | CCSA | NEM | dSNE | Ours |
|---|---|---|---|---|---|---|---|
| 10 | $40.55 \pm 1.14$ | $53.98 \pm 1.47$ | $60.64 \pm 1.36$ | $48.43 \pm 1.46$ | $\underline{61.88 \pm 1.33}$ | $60.54 \pm 1.05$ | $\mathbf{62.90 \pm 1.13}$ |
| 15 | $38.89 \pm 1.24$ | $60.14 \pm 1.35$ | $62.76 \pm 0.77$ | $47.43 \pm 0.99$ | $\mathbf{64.66 \pm 0.94}$ | $63.71 \pm 1.20$ | $\underline{64.39 \pm 1.07}$ |
| 20 | $35.12 \pm 1.12$ | $62.62 \pm 0.71$ | $\underline{66.26 \pm 0.73}$ | $49.55 \pm 0.86$ | $65.80 \pm 0.80$ | $65.41 \pm 0.98$ | $\mathbf{66.55 \pm 1.22}$ |

Table 4: Classification accuracy for zero-shot DA task. In each column, we consider a different proportion (number) of "cold" classes.

| | 70% (7) | 50% (5) | 20% (2) |
|---|---|---|---|
| CCSA | $72.67 \pm 5.68$ | $75.29 \pm 2.58$ | $77.19 \pm 5.02$ |
| d-SNE | $\underline{73.41 \pm 4.33}$ | $\underline{78.09 \pm 4.23}$ | $82.99 \pm 2.83$ |
| Ours | $\mathbf{77.24 \pm 2.25}$ | $\mathbf{81.36 \pm 1.26}$ | $\mathbf{84.00 \pm 2.35}$ |

training set contains only images that are associated with 1-hot output vectors, i.e., they consist of a single digit. For the training set from the target domain, we follow a similar approach. We randomly pick 32 labels, and for each label we randomly generate 3 training datapoints. An illustration of these domains is presented in Figure 3(a).

We repeat this experiment 10 times and report in Table 5 three evaluation metrics for this multi-label classification: F1-Score, AUC, and the Hamming distance between the predicted indicator vectors and the true indicator vectors. We see that our approach obtained superior results compared to CCSA and d-SNE. We note that for fair comparison we adapted both CCSA and d-SNE to multi-label classification. See Appendix C for details.

In the third experiment, we consider a colored version of the "Digits" tasks. Specifically, we replace the original single-digit grayscale images with colored versions of multi-digit images. Each image is a $\left(1 + \lceil \frac{N_c}{10} \rceil\right)$-digit number, where $N_c$ denotes the number of colors in the color palette. For example, for $N_c = 3$, we get 2-digit images. The digits in the image are randomly sampled from the corresponding domain (MNIST or USPS). The color of each image is determined by the $\lceil \frac{N_c}{10} \rceil$ most significant digits in the image, and the label is set according to the presented number in the image. The resulting dataset consists of $10 \cdot N_c$ classes, divided into $N_c$ clusters dominated by the color of the images. In Figure 3(b) we present an illustrative t-SNE embedding Van der Maaten & Hinton (2008) of a colored version of the MNIST dataset for $N_c = 3$. Observing Figure 3(b), we can see that the input space geometry is indeed dominated by the colors of the images. We repeat the same experiment as in Section 6.1 but for a DA task from colored MNIST to colored USPS. We expect our approach to capture the geometry of the output space, in contrast to the competing methods that map input data of the same class to a single point. This may be conveyed in our approach by mapping input data with the same color but different digits to nearby locations in the embedded space, thereby facilitating fewer errors in predicting the true color. Therefore, in addition to the accuracy measure, we report the proportion of partial errors, where the color of the predicted classes is different than the color of the true class. The results are presented in the table in Table 6.

In the upper part of the table, we show the obtained accuracy for $N_c = 10, 20, 30$. We see that the geometric-aware methods (ours and d-SNE) obtain superior results compared to the contrastive-based approach (CCSA). At the lower part of Table 6 we present the proportion of out-of-color (OOC) errors. We see that the geometric-aware methods (ours and d-SNE) manage to yield significantly lower OOC errors.

## 6.3 Regression

We demonstrate the universality of our method and apply it to regression tasks. In contrast to classification tasks, to the best of our knowledge, there is no definitive benchmark for SDA in regression tasks. Here, we consider two datasets that were used in two recent SDA methods for regression.

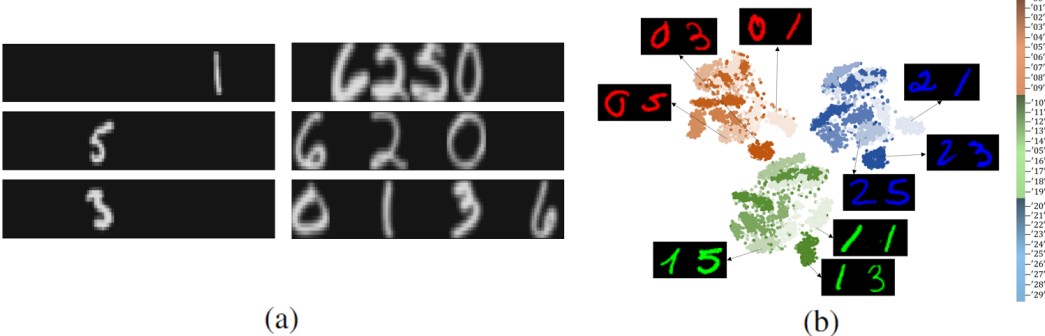

Figure 3: (a) Illustration of the images from the multi-label classification task. The images on the left (right) side are associated with the source (target) domain. (b) An illustrative t-SNE embeddings of the raw datapoints from a colored MNIST dataset for $N_c = 3$. Each image is a 2-digit number, where the color is encoded in the tens digit.

Table 5: F1-Score, AUC (higher is better), and Hamming distance (lower is better) for the multi-label DA task.

|  | F1-Score ($\uparrow$) | AUC ($\uparrow$) | Hamming ($\downarrow$) |
|---|---|---|---|
| CCSA | $0.22 \pm 0.25$ | $0.55 \pm 0.16$ | $0.46 \pm 0.10$ |
| d-SNE | $0.69 \pm 0.15$ | $0.73 \pm 0.10$ | $0.31 \pm 0.08$ |
| Ours | $\mathbf{0.74 \pm 0.15}$ | $\mathbf{0.80 \pm 0.16}$ | $\mathbf{0.26 \pm 0.15}$ |

The first experiment is taken from de Mathelin et al. (2021) and is based on the CityCam vehicle counting dataset Zhang et al. (2017). This dataset consists of images acquired by traffic cameras, where each image is annotated with a label indicating the number of vehicles in a certain range of interest. We follow the same experimental protocol as in de Mathelin et al. (2021), where the authors consider images from 4 cameras: two located on a highway and two located at an intersection. The images from one of the intersections are considered as the target domain, and the rest are considered as the source domain. For more details, see Appendix C. We consider $20, 50, 100$ and $200$ samples from the target domain and compute the obtained mean-absolute-error (MAE). In Table 7, we present our results along with the results from de Mathelin et al. (2021). We see that our approach outperforms classical approaches such as TrAdaB Pardoe & Stone (2010), KLIEP Sugiyama et al. (2007), and KMM Huang et al. (2006), while obtaining competitive results compared to the recent WANN de Mathelin et al. (2021), especially when fewer samples from the target domain are available.

The second experiment is taken from Teshima et al. (2020), and it is based on a gasoline consumption dataset (William (2008), p.284, Example 9.5). This dataset consists of tabular data describing gasoline usage in 18 of the OECD countries over 19 years. We follow the same experimental protocol as in Teshima et al. (2020), where the experiment is repeated 18 times. Each time, one country is considered as the target domain, and the rest are considered as the source domain. The evaluation metric considered in Teshima et al. (2020) is a normalized version of the mean square error (NMSE). For more details, see Appendix C. In Table 8, we present our results along with the results from Teshima et al. (2020). For convenience, in addition to the evaluated metric from Teshima et al. (2020), on the right side of Table 8, we present the performance ranking obtained by each method for each target country. We see that our approach performs better for most of the domains as well as on average.

## 7    Conclusions

In this paper, we proposed a new approach for SDA. We presented an upper bound of the generalization error in the target domain, which extends the setting considered by Ben-David et al. (2010). Based on

Table 6: Adaptation results of the colored MNIST to the colored USPS. In each column, we consider a different number of colors $N_c$. In the upper part, we show the accuracy obtained by each method, and in the lower part, we present the proportion of out-of-color errors.

|  |  | 10 | 20 | 30 |
|---|---|---|---|---|
| Acc (↑) | CCSA | $\underline{77.49 \pm 0.84}$ | $74.90 \pm 2.53$ | $68.86 \pm 1.76$ |
|  | d-SNE | $75.55 \pm 1.72$ | $\underline{75.22 \pm 1.25}$ | $\underline{71.71 \pm 2.81}$ |
|  | Ours | $\mathbf{78.54 \pm 1.74}$ | $\mathbf{80.45 \pm 0.69}$ | $\mathbf{80.71 \pm 1.14}$ |
| OOC (↓) | CCSA | $\underline{0.67 \pm 0.52}$ | $7.95 \pm 1.73$ | $25.07 \pm 4.68$ |
|  | d-SNE | $0.73 \pm 0.17$ | $\underline{5.33 \pm 1.41}$ | $\underline{16.14 \pm 3.70}$ |
|  | Ours | $\mathbf{0.23 \pm 0.17}$ | $\mathbf{1.32 \pm 0.45}$ | $\mathbf{5.67 \pm 0.92}$ |

Table 7: MAE for the CityCam vehicle counting dataset. In each row, we consider a different number of samples from the target domain.

|  | TrAdaB. | KLIEP | KMM | WANN | Ours |
|---|---|---|---|---|---|
| 20 | $3.30 \pm 0.18$ | $3.05 \pm 0.48$ | $\underline{2.79 \pm 0.16}$ | $2.79 \pm 0.22$ | $\mathbf{2.67 \pm 0.30}$ |
| 50 | $2.96 \pm 0.13$ | $2.60 \pm 0.15$ | $2.52 \pm 0.10$ | $\underline{2.48 \pm 0.08}$ | $\mathbf{2.41 \pm 0.19}$ |
| 100 | $2.52 \pm 0.16$ | $2.34 \pm 0.1$ | $2.32 \pm 0.07$ | $\underline{2.26 \pm 0.04}$ | $\mathbf{2.21 \pm 0.32}$ |
| 200 | $2.28 \pm 0.11$ | $2.07 \pm 0.07$ | $2.06 \pm 0.09$ | $\mathbf{1.98 \pm 0.07}$ | $\underline{2.05 \pm 0.23}$ |

Table 8: Results for the gasoline dataset. In each row, a different country is considered the target domain. On the left side, we present the Normalized MSE as presented in Teshima et al. (2020). On the right side, we present the ranking of the methods (lower is better).

| Target Country | T-Only | S-Only | S+T | CMTra | Ours | Target Country | T-Only | S-Only | S+T | CMTra | Ours |
|---|---|---|---|---|---|---|---|---|---|---|---|
| AUT | $5.88 \pm 1.60$ | $9.67 \pm 0.57$ | $9.84 \pm 0.62$ | $5.39 \pm 1.86$ | $\mathbf{2.89 \pm 1.47}$ | AUT | 3 | 4 | 5 | 2 | 1 |
| BEL | $10.70 \pm 7.50$ | $8.19 \pm 0.68$ | $9.48 \pm 0.91$ | $7.94 \pm 2.19$ | $\mathbf{3.36 \pm 1.48}$ | BEL | 5 | 3 | 4 | 2 | 1 |
| CAN | $5.16 \pm 1.36$ | $157.74 \pm 8.83$ | $156.65 \pm 10.69$ | $\mathbf{3.84 \pm 0.98}$ | $15.29 \pm 4.03$ | CAN | 2 | 5 | 4 | 1 | 3 |
| DNK | $3.26 \pm 0.61$ | $30.79 \pm 0.93$ | $28.12 \pm 1.67$ | $\mathbf{3.23 \pm 0.63}$ | $3.45 \pm 1.97$ | DNK | 2 | 5 | 4 | 1 | 3 |
| FRA | $2.79 \pm 1.10$ | $4.67 \pm 0.41$ | $3.05 \pm 0.11$ | $1.92 \pm 0.66$ | $\mathbf{1.20 \pm 0.67}$ | FRA | 3 | 5 | 4 | 2 | 1 |
| DEU | $16.99 \pm 8.04$ | $229.65 \pm 9.13$ | $210.59 \pm 14.99$ | $\mathbf{6.71 \pm 1.23}$ | $30.71 \pm 4.54$ | DEU | 2 | 5 | 4 | 1 | 3 |
| GRC | $3.80 \pm 2.21$ | $5.30 \pm 0.90$ | $5.75 \pm 0.68$ | $3.55 \pm 1.79$ | $\mathbf{1.85 \pm 1.42}$ | GRC | 3 | 4 | 5 | 2 | 1 |
| IRL | $\mathbf{3.05 \pm 0.34}$ | $135.57 \pm 5.64$ | $12.34 \pm 0.58$ | $4.35 \pm 1.25$ | $82.22 \pm 1.46$ | IRL | 1 | 5 | 3 | 2 | 4 |
| ITA | $13.00 \pm 4.15$ | $35.29 \pm 1.83$ | $39.27 \pm 2.52$ | $14.05 \pm 4.81$ | $\mathbf{3.64 \pm 2.19}$ | ITA | 2 | 4 | 5 | 3 | 1 |
| JPN | $10.55 \pm 4.67$ | $8.10 \pm 1.05$ | $8.38 \pm 1.07$ | $12.32 \pm 4.95$ | $\mathbf{0.48 \pm 0.61}$ | JPN | 4 | 2 | 3 | 5 | 1 |
| NLD | $3.75 \pm 0.80$ | $\mathbf{0.99 \pm 0.06}$ | $0.99 \pm 0.05$ | $3.87 \pm 0.79$ | $80.30 \pm 0.30$ | NLD | 3 | 1 | 2 | 4 | 5 |
| NOR | $2.70 \pm 0.51$ | $1.86 \pm 0.29$ | $\mathbf{1.63 \pm 0.11}$ | $2.82 \pm 0.73$ | $80.66 \pm 0.59$ | NOR | 3 | 2 | 1 | 4 | 5 |
| ESP | $5.18 \pm 1.05$ | $5.17 \pm 1.14$ | $4.29 \pm 0.72$ | $6.09 \pm 1.53$ | $\mathbf{0.54 \pm 0.48}$ | ESP | 4 | 3 | 2 | 5 | 1 |
| SWE | $6.44 \pm 2.66$ | $2.48 \pm 0.23$ | $2.02 \pm 0.21$ | $5.47 \pm 2.63$ | $\mathbf{1.42 \pm 1.05}$ | SWE | 5 | 3 | 2 | 4 | 1 |
| CHE | $3.51 \pm 0.46$ | $43.59 \pm 1.77$ | $7.48 \pm 0.49$ | $2.90 \pm 0.37$ | $\mathbf{1.93 \pm 1.23}$ | CHE | 3 | 5 | 4 | 2 | 1 |
| TUR | $1.65 \pm 0.47$ | $1.22 \pm 0.18$ | $\mathbf{0.91 \pm 0.09}$ | $1.06 \pm 0.15$ | $6.30 \pm 9.98$ | TUR | 4 | 3 | 1 | 2 | 5 |
| GBR | $5.95 \pm 1.86$ | $15.92 \pm 1.02$ | $10.05 \pm 1.47$ | $\mathbf{2.66 \pm 0.57}$ | $5.07 \pm 1.81$ | GBR | 3 | 5 | 4 | 1 | 2 |
| USA | $4.98 \pm 1.96$ | $21.53 \pm 3.30$ | $12.28 \pm 2.52$ | $\mathbf{1.60 \pm 0.42}$ | $18.41 \pm 3.20$ | USA | 2 | 5 | 3 | 1 | 4 |
| #Best | 1 | 1 | 2 | 5 | **9** | Mean | 3 | 3.83 | 3.33 | 2.44 | **2.39** |

this bound, we formulated the task of SDA as a new optimization problem. We proposed to solve this optimization problem using a Siamese neural network. Experimental results demonstrate the effectiveness and broad applicability of the proposed approach for various classification and regression tasks.

We hope this work will spark interest in employing the principles of Ben-David et al. (2010) to SDA tasks. In future work, we will extend the proposed approach for semi-supervised domain adaptation. This is a natural extension of the proposed approach that follows the decoupling between the input and labels distributions in the derived bound.

## Acknowledgments and Disclosure of Funding

We thank the editor and the reviewers for their important comments and suggestions. The work of OK and RT was supported by the European Union's Horizon 2020 research and innovation programme under grant agreement No. 802735-ERC-DIFFOP. RT acknowledges the support of the Schmidt Career Advancement Chair in AI.

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

# Appendix

## A  Proof for Theorem 4.1

*Proof.* Recall the definition for the error of a hypothesis: $\epsilon_{\mathcal{P}}(h) \triangleq \mathbb{E}_{(x,y)\sim\mathcal{P}} \ell(h(x), y)$, which we can recast the error as $\epsilon_{\mathcal{P}}(h) = \mathbb{E}_{x\in\mathcal{Q}} \mathbb{E}_{y\in\mathcal{P}_x} \ell(h(x), y)$. In addition, let $Q^v(\cdot)$ denote the density function of the marginal distribution $\mathcal{Q}^v$ for $v \in \{S, T\}$. Then, we have:

$$\epsilon_{\mathcal{P}^T}(h) = \epsilon_{\mathcal{P}^T}(h) + \epsilon_{\mathcal{P}^S}(h) - \epsilon_{\mathcal{P}^S}(h) + \mathbb{E}_{x\in\mathcal{Q}^S} \mathbb{E}_{y\in\mathcal{P}_x^T} \ell(h(x), y) - \mathbb{E}_{x\in\mathcal{Q}^S} \mathbb{E}_{y\in\mathcal{P}_x^T} \ell(h(x), y) \tag{6}$$

$$\leq \epsilon_{\mathcal{P}^S}(h) + \left| \mathbb{E}_{x\in\mathcal{Q}^S} \mathbb{E}_{y\in\mathcal{P}_x^T} \ell(h(x), y) - \epsilon_{\mathcal{P}^S}(h) \right| + \left| \epsilon_{\mathcal{P}^T}(h) - \mathbb{E}_{x\in\mathcal{Q}^S} \mathbb{E}_{y\in\mathcal{P}_x^T} \ell(h(x), y) \right| \tag{7}$$

$$= \epsilon_{\mathcal{P}^S}(h) + \left| \mathbb{E}_{x\in\mathcal{Q}^S} \mathbb{E}_{y\in\mathcal{P}_x^T} \ell(h(x), y) - \mathbb{E}_{x\in\mathcal{Q}^S} \mathbb{E}_{y'\in\mathcal{P}_x^S} \ell(h(x), y') \right| +$$

$$\left| \mathbb{E}_{x\in\mathcal{Q}^T} \mathbb{E}_{y\in\mathcal{P}_x^T} \ell(h(x), y) - \mathbb{E}_{x\in\mathcal{Q}^S} \mathbb{E}_{y\in\mathcal{P}_x^T} \ell(h(x), y) \right| \tag{8}$$

$$= \epsilon_{\mathcal{P}^S}(h) + \left| \mathbb{E}_{\substack{x\in\mathcal{Q}^S \\ y\sim\mathcal{P}_x^S \\ y'\sim\mathcal{P}_x^T}} \ell(h(x), y) - \ell(h(x), y') \right| +$$

$$\left| \mathbb{E}_{x\in\mathcal{Q}^T} \mathbb{E}_{y\in\mathcal{P}_x^T} \ell(h(x), y) - \mathbb{E}_{x\in\mathcal{Q}^S} \mathbb{E}_{y\in\mathcal{P}_x^T} \ell(h(x), y) \right| \tag{9}$$

$$\leq \epsilon_{\mathcal{P}^S}(h) + \mathbb{E}_{\substack{x\in\mathcal{Q}^S \\ y\sim\mathcal{P}_x^S \\ y'\sim\mathcal{P}_x^T}} |\ell(h(x), y) - \ell(h(x), y')| +$$

$$\int_{x\in\mathcal{X}} \left| Q^S(x) \mathbb{E}_{y\sim\mathcal{P}_x^T} \ell(h(x), y) - Q^T(x) \mathbb{E}_{y\sim\mathcal{P}_x^T} \ell(h(x), y) \right| dx \tag{10}$$

$$= \epsilon_{\mathcal{P}^S}(h) + \mathbb{E}_{\substack{x\in\mathcal{Q}^S \\ y\sim\mathcal{P}_x^S \\ y'\sim\mathcal{P}_x^T}} |\ell(h(x), y) - \ell(h(x), y')| +$$

$$\int_{x\in\mathcal{X}} \left| Q^S(x) - Q^T(x) \right| \mathbb{E}_{y\sim\mathcal{P}_x^T} \ell(h(x), y) dx \tag{11}$$

$$\leq \epsilon_{\mathcal{P}^S}(h) + \mathbb{E}_{\substack{x\in\mathcal{Q}^S \\ y\sim\mathcal{P}_x^S \\ y'\sim\mathcal{P}_x^T}} \ell(y, y') + M \int_{x\in\mathcal{X}} \left| Q^S(x) - Q^T(x) \right| dx \tag{12}$$

$$= \epsilon_{\mathcal{P}^S}(h) + \chi^S + M d(\mathcal{Q}^T, \mathcal{Q}^S), \tag{13}$$

where the transition from equation 11 to equation 12 is due to the assumption: $|\ell(x, y) - \ell(x, z)| \leq |\ell(y, z)|$ and the upper bound of the expected loss function $M$. Repeating the same derivation, but replacing $\epsilon_{\mathcal{P}^S}(h)$ by $\epsilon_{\mathcal{P}^T}(h)$ in the r.h.s of equation 6 results in a similar bound: $\epsilon_{\mathcal{P}^T}(h) \leq \epsilon_{\mathcal{P}^S}(h) + \chi^T + M d(\mathcal{Q}^T, \mathcal{Q}^S)$, implying that:

$$\epsilon_{\mathcal{P}^T}(h) \leq \epsilon_{\mathcal{P}^S}(h) + M d(\mathcal{Q}^S, \mathcal{Q}^T) + \min\{\chi^S, \chi^T\}.$$

$\square$

## B  Baselines Reproducibility on Benchmarks Dataset

In our experimental study, we faced difficulties reproducing the results on the benchmark datasets of "Digits" and "Office" of the two most recognized methods for SDA: CCSA Motiian et al. (2017b) and dSNE Xu et al. (2019). These difficulties stem from the following reasons:

- **Known implementation issues:** When considering few-shot learning scenarios the randomly picked training datapoints from the target domain can highly affect the results. Therefore, a common practice is to repeat each experiment multiple times. This is indeed the practice that appears in the source code published by the authors of CCSA [1]. However, in their code, the model parameters are not initialized between each experiment. As a consequence, the model accumulates knowledge from previous experiments. Note that this is a known issue [2], which the authors have not yet addressed. Similarly, the authors of dSNE published their code [3]. However, we, among others, could not reproduce their results[4].

- **Backbone inconsistencies:** In recent years, many DA methods are based on Siamese neural networks. The choice of the backbone, its architecture, and whether it is pre-trained or not, highly affect the results (for example, see the bottom line in Table I in Hedegaard et al. (2021)). Therefore, in order to fairly compare DA techniques, it is crucial to perform the evaluation with the same backbone. We note that in the literature, there are works that do not follow this guideline. For example, in dSNE, for the "Digits" datasets, instead of using the same backbone as in Motiian et al. (2017b;a), a close look at the code reveals that LeNet++ was used Wen et al. (2016).

- **Datasets:** At first glance, the protocol for preparing the benchmark datasets is straightforward, as described in Section 6, and in more detail in Appendix C. However, the following aspects are often overlooked:
  - **Augmentations.** Datasets augmentations can highly improve the model's ability to generalize, especially when evaluating the ability of a model that was trained on catalog images (Amazon) to classify images from the wild (DLSR or Webcam). Similar to the backbone, a fair comparison between methods should rely on the same augmentations protocol.
  - **Partitioning.** Although the train-test partitioning is clearly defined, it is not clear how to construct the validation set. For example, in the extreme case of 1 sample per target class, choosing the validation set is not straightforward. Inspecting the code of recently published papers shows that the prevalent practice is to use the test set as validation. The absence of a validation set introduces a risk for model selection, especially when systematic hyperparameter tuning is conducted. This concern was recently raised by Hedegaard et al. (2021), where the authors proposed a rectified experimental protocol and released a Python package for benchmark datasets [5][6]. This protocol is yet to be considered as a standard, however, in our experiments, we adopted their protocol and we wish to encourage others to do so as well. We note that this protocol only mitigates the difficulties that stem from the inability to properly define a validation set. Although the target datapoints in the validation set are not explicitly introduced to the model during training, they do affect the model's hyperparameters. Therefore, in order to perform a fair comparison between different models is it important to ensure that the same computational budget was used in the hyperparameter tuning of each method.

- **Hyperparameters inconsistency:** Similarly to the choice of the backbone, the selection of the hyperparameters can highly affect the model's performance. Therefore, in order to conduct a fair comparison between different models, it is important to ensure that the same hyperparameters are tuned in each experiment. Otherwise, especially in the absence of a validation set, the experiment

---

[1]https://github.com/samotiian/CCSA
[2]https://github.com/samotiian/CCSA/issues/10
[3]https://github.com/aws-samples/d-SNE
[4]https://github.com/aws-samples/d-SNE/issues/13
[5]https://pypi.org/project/office31
[6]https://pypi.org/project/mnistusps

results might reflect the quality of the hyperparameters tuning procedure rather than the quality of the evaluated model. Many works, however, do not follow this practice. For example, in Hedegaard et al. (2021), the set of hyperparameters employed by the model was significantly larger than the ones considered in Motiian et al. (2017b;a); Xu et al. (2019). Moreover, the hyperparameters tuning protocol conducted by Hedegaard et al. (2021) requires large computational resources.

Therefore, when considering the results obtained by a specific method in a specific experiment, we see a wide range of values reported in different papers. The results for CCSA Motiian et al. (2017b) and dSNE Xu et al. (2019), as reported by different papers for the MNIST to USPS experiment are presented in Table 9. We see that although the same method was evaluated on the same benchmark, different papers report significantly different results. A similar inconsistency was observed in the experiments related to the "Office" dataset, as can be seen in Table 10. Note that the implementation of CCSA in Morsing et al. (2021) and in Hedegaard et al. (2021) deviates from the original one proposed in Motiian et al. (2017b) and includes the explicit error term on the target domain.

In order to cope with the inability to rely on prior results, we reproduced the baseline results using the same pipeline employed for all the methods.

Table 9: Experimental results for the MNIST to USPS experiment as reported by different papers (Table 1 in this paper). The results in the upper part are related to CCSA and the results in the lower part are related to dSNE.

| | Reported by | 1 | 3 | 5 | 7 |
|---|---|---|---|---|---|
| CCSA | Motiian et al. (2017b) (Authors) | 85.0 | 90.1 | 92.4 | 92.9 |
| | Reproduced by us | $85.11 \pm 1.23$ | $87.51 \pm 1.15$ | $89.60 \pm 0.94$ | $90.35 \pm 0.71$ |
| | Morsing et al. (2021) | $75.6 \pm 2.1$ | $85.0 \pm 1.4$ | $87.8 \pm 0.7$ | $89.1 \pm 0.7$ |
| | Hedegaard et al. (2021) | $89.1 \pm 1.1$ | $91.2 \pm 0.9$ | $93.8 \pm 0.4$ | $94.3 \pm 0.4$ |
| dSNE | Xu et al. (2019) (Authors) | 92.9 | 93.55 | 95.13 | 96.13 |
| | Reproduced by us | $84.39 \pm 1.42$ | $89.04 \pm 1.14$ | $89.42 \pm 1.05$ | $91.34 \pm 0.88$ |
| | Morsing et al. (2021) | $69.0 \pm 1.7$ | $80.4 \pm 1.7$ | $86.1 \pm 0.9$ | $87.7 \pm 0.9$ |
| | Hedegaard et al. (2021) | $88.3 \pm 1.7$ | $91.4 \pm 1.2$ | $93.1 \pm 0.5$ | $93.6 \pm 0.6$ |

Table 10: Experimental results for the "Office" experiment as reported by different papers (Table 2 in this paper). The results in the upper part are related to CCSA and the results in the lower part are related to dSNE.

| | Reported by | $\mathcal{A} \to \mathcal{D}$ | $\mathcal{A} \to \mathcal{W}$ | $\mathcal{D} \to \mathcal{A}$ | $\mathcal{D} \to \mathcal{W}$ | $\mathcal{W} \to \mathcal{A}$ | $\mathcal{W} \to \mathcal{D}$ |
|---|---|---|---|---|---|---|---|
| CCSA | Motiian et al. (2017b) (Authors) | $89.0 \pm 1.2$ | $88.2 \pm 1.0$ | $71.8 \pm 0.5$ | $96.4 \pm 0.8$ | $72.1 \pm 1.0$ | $97.6 \pm 0.4$ |
| | Reproduced by us | $70.41 \pm 1.52$ | $71.72 \pm 1.27$ | $59.33 \pm 1.37$ | $94.37 \pm 0.86$ | $57.26 \pm 1.07$ | $94.53 \pm 1.21$ |
| | Morsing et al. (2021) | $84.8 \pm 2.1$ | $87.5 \pm 1.5$ | $66.5 \pm 1.9$ | $97.2 \pm 0.7$ | $64.0 \pm 1.6$ | $98.6 \pm 0.4$ |
| | Hedegaard et al. (2021) | $86.4 \pm 2.5$ | $84.5 \pm 2.1$ | $65.5 \pm 1.2$ | $97.5 \pm 0.9$ | $60.8 \pm 1.5$ | $98.4 \pm 1.0$ |
| dSNE | Xu et al. (2019) (Authors) | $91.4 \pm 0.23$ | $90.1 \pm 0.07$ | $71.1 \pm 0.18$ | $97.1 \pm 0.07$ | $71.7 \pm 0.42$ | $97.5 \pm 0.24$ |
| | Reproduced by us | $83.50 \pm 2.25$ | $84.58 \pm 1.52$ | $65.52 \pm 0.99$ | $94.06 \pm 0.77$ | $64.59 \pm 1.33$ | $96.35 \pm 1.01$ |
| | Morsing et al. (2021) | $86.5 \pm 2.5$ | $88.7 \pm 1.9$ | $65.9 \pm 1.1$ | $97.6 \pm 0.7$ | $63.9 \pm 1.2$ | $99.0 \pm 0.5$ |
| | Hedegaard et al. (2021) | $84.7 \pm 1.3$ | $82.3 \pm 2.4$ | $65.1 \pm 0.9$ | $98.2 \pm 0.4$ | $59.9 \pm 1.6$ | $99.7 \pm 0.4$ |

## C   Technical Details and Additional Results

In this section, we provide more details and additional results to the experimental study in Section 6. Our code is publicly available here.

### C.1   Benchmark datasets

In these experiments, we follow the experimental setting from Motiian et al. (2017b); Xu et al. (2019). The data partitioning is carried out according to the rectified experimental protocol suggested in Hedegaard

et al. (2021) using their released Python packages [7] [8]. The hyperparameters set includes the mini-batch size, optimizer selection (Adam Kingma & Ba (2014) or SGD), learning rate, weight-decay, and the number of training epochs. Hyperparameters tuning was carried out manually based on the validation set for each experiment, and then, the same set of hyperparameters was used for all the methods. The selected hyperparameters for each experiment appear in our published code. The training loss in these experiments is the binary cross-entropy (BCE) loss. The CDCA term is applied by representing the labels as 1-hot vectors in $\mathbb{R}^C$, where $C$ denotes the cardinality of the labels space.

### C.1.1 "Digits" experiments

In this setting, each experiment is repeated 10 times. In each experiment, 200 samples per class from the source domain and $X$ samples per class from the target domain are randomly selected as the training set, where $X \in \{1, 3, 5, 7\}$. The images from the MNIST dataset are resized to fit the size of the images from the USPS dataset. No data augmentation was applied. The network architecture consists of two convolutional layers with a kernel size of $3 \times 3$ and 32 filters followed by max-pooling layers and 2 fully connected layers of sizes 120 and 84. We note that this architecture deviates from the architecture reported in Motiian et al. (2017b), but it is taken from their published source code [9] (see lines 94-98). We further emphasize that we used the same architecture for all the evaluated methods.

### C.1.2 "Office" and "VisDA-C" experiments

In this setting, each experiment is repeated 5 times. In each experiment, the samples for each class are sampled according to the quantities described in Section 6. In line with the experimental protocol from Motiian et al. (2017b); Xu et al. (2019), for the "Office" tasks we perform the default data augmentations used in the Python package TLlib [10]. These augmentations are based on a random resized crop of size $224 \times 224$. The network architecture consists of the convolutional layers of a VGG-16 model Kobayashi et al. (2015) pre-trained on ImageNet Russakovsky et al. (2015), followed by 2 fully connected layers of sizes 1024 and 128.

### C.2 Zero-shot, multi-label, and colored "Digits" experiment

For the zero-shot experiment, we used the same architecture as in the "Digits" experiment. For the multi-label and colored experiment, we made subtle modifications to support the size of the modified image. Specifically, for the multi-label experiment, we modified the fully-connected layer located after the max-pooling layer from $1152 \times 120$ to $10368 \times 120$. For the colored experiment, we changed the first convolutional layer to support 3 (RGB) input channels. The training loss in the zero-shot and the colored digits experiments is the BCE loss, and in the multi-label experiment it is the sum of the BCE loss applied to each slot. The CDCA term is implemented by representing the labels as vectors in $\mathbb{R}^C$, where $C$ denotes the cardinality of the label space. In the zero-shot and the colored digits experiments, these vectors are 1-hot, whereas in the multi-label experiment, they may contain any binary representation.

### C.3 CityCam experiment

We followed the experiment protocol from de Mathelin et al. (2021).The CityCam dataset is taken from Zhang et al. (2017) and is publicly available[11]. Four cameras from the dataset were selected: "253", "511", "572", and "495". Cameras "253", "511", and "572" are considered as the source domain, while camera "495" is considered as the target domain. Each experiment was repeated 10 times. In each experiment, $X$ datapoints from the target domain were randomly selected for training, while the rest were kept for testing. All the datapoints from the source domain are used for training. The images acquired by each camera were pre-processed using a pre-trained ResNet50 model.The pre-processing code can be found here [12], and the

---

[7] https://pypi.org/project/mnistusps
[8] https://pypi.org/project/office31
[9] https://github.com/samotiian/CCSA/blob/master/Initialization.py
[10] https://github.com/thuml/Transfer-Learning-Library
[11] https://www.citycam-cmu.com/dataset
[12] https://github.com/antoinedemathelin/wann/blob/master/notebooks/CityCam_preprocessing.ipynb

processed dataset can be found here [13]. We used the same network architecture as in de Mathelin et al. (2021), which consists of a multi-layer perceptron with two hidden layers of size 100 and 10. The training loss is the MSE loss.

### C.4 Gasoline experiment

We followed the experiment protocol from Teshima et al. (2020). The Gasoline dataset is publicly available here [14], its parsed version is available here [15]. The dataset consists of gasoline usage in 18 of the OECD countries over 19 years. Each record in the dataset describes the gasoline consumption in a certain year and country and contains four variables: per-capita income, gasoline price, stock of cars per capita, and gasoline consumption per car. The first three are considered as the predictor variable and the latter is considered as the label. We follow the same procedure as in Teshima et al. (2020), where the time-series structure (i.e., temporal order) is ignored, and instead the data is considered as i.i.d. samples for each country. For each country, we perform the following procedure. We randomly pick 6 datapoints from their associated records to create the target training dataset, while the rest of its associated datapoints are used for testing. The source training dataset consists of all the datapoints from the rest of the countries. The training loss is the MSE loss. We use the same evaluation metric as in Teshima et al. (2020), which is the MSE distance normalized by the closest label in the target set (including the test datapoints). We repeat this procedure 10 times with different train-test splits of target domain data.

## D Implementation, Computational Complexity and Limitations

Incorporating geometrical considerations typically requires the ability to process large mini-batches (see for example Xu et al. (2019)). Indeed, traditional methods for UDA, such as DDC Tzeng et al. (2014), DAN Long et al. (2015), and CORAL Sun et al. (2017) require large mini-batches. However, this limitation can be mitigated by considering adversarial approaches for UDA, such as ADDA Tzeng et al. (2017). In addition to the covariate shift, which is addressed using UDA methods, our approach relies on the approximation of the CDCA term. Since we approximate this term using non-parametric kernel regression, its resolution is also limited by the size of the mini-batch. This challenge will be addressed in future work by implementing the CDCA term parametrically. More specifically, following the recent advances in UDA and the emergence of adversarial methods, we plan to use an adversarial mechanism to sample from the cross-domain, rather than generate samples using the non-parametric kernel regression.

---

[13]https://github.com/antoinedemathelin/wann/tree/master/dataset
[14]https://bcs.wiley.com/he-bcs/Books?action=resource&bcsId=4338&itemId=1118672321&resourceId=13452
[15]https://github.com/takeshi-teshima/few-shot-domain-adaptation-by-causal-mechanism-transfer/tree/master/experiments/icml2020/data/gaso

