# OpenReview forum: "Supervised Domain Adaptation Based on Marginal and Conditional Distributions Alignment"
_TMLR — Accepted by TMLR_

### Review · Reviewer_DCB5 · 2024-04-01

**Summary Of Contributions:**

The paper studies the problem of supervised domain adaptation. It shows an upper bound for the error. The paper then utilizes the upper bound as a training objective and proposes training a Siamese network for the supervised domain adaptation problem. The paper shows that the proposed learning method can achieve certain improvements, especially for zero- or few-shot tasks.

**Audience:**

Yes

**Claims And Evidence:**

Yes

**Requested Changes:**

Please address the weaknesses part.

**Strengths And Weaknesses:**

Strengths:
1. The paper shows an upper bound for the error of supervised domain adaption under mild assumptions.

2. The paper shows that such an upper bound can be utilized as a training objective for a Siamese network to get good performance in certain settings for supervised domain adaption.

Weaknesses:
1. The tightness of the upper bound is unknown. Indeed, there could be some steps during the derivations (e.g., eq 10 and 12) contributing to large bounding factors. Therefore, it is not clear that such an upper bound could be a good objective to optimize either in theory or in practice.

2. The final optimization approach is quite similar to contrastive learning, but its connections and differences are not well discussed.

3. The proposed method does not show very consistent results across settings. For example, why does it perform better for the M to U adaptation in Table 1 but not vice versa? Also, the network structures used for the experiments are a bit small, considering the recent literature. Recent work on these datasets typically uses ResNet-50 or similar architectures.

---

> ### Author Response · Authors · 2024-04-24
> **Respones to Reviewer DCB5 - Part 1**
>
> We wish to thank the reviewer for these comments that helped us improve the manuscript. We have modified the manuscript to address the comments raised. Below is a list of the point-by-point responses and all the modifications we have made.
>
> 1. The transition between eq. 11 and eq. 12 in the derivation of the bound in Theorem 4.1 relies on two assumptions.  The first assumption requires that the loss function will satisfy: $|\ell(x,y)-\ell(x,z)|\leq |\ell(y,z)|$. For example, due to the reverse triangle inequality, this assumption holds for the $L_1$ loss function. Cross-entropy (CE) does not generally satisfy this assumption. However, in certain cases involving densities that result from CE minimization, it does (See Properties 10, 12, and 13 in [1]). The second assumption requires that the expectation of the loss function, denoted by $M$, is bounded. For example, for the $0-1$ loss function, $M$ is bounded by 1. We acknowledge that, although $\epsilon_T(h)$ is bounded by $M$, this is a very loose upper bound. In our proposed loss term, $M$ is multiplied by $d(\widetilde{\mathcal{Q}}^{\text{S}},\widetilde{\mathcal{Q}}^{\text{T}})$ which is minimized during the training process.
> We wish to conclude this answer by stating that our proposed method is only motivated by the upper bound in Theorem 4.1. In practice, since minimizing the expected $0-1$ loss is typically intractable, we use CE as a surrogate loss function instead. In these cases, $M$ is unbounded, however, our experimental study shows that our method remains applicable and useful.
> Following your comment, we have modified the manuscript and included these remarks (highlighted in red in Section 4).
>
>
> 2. As you mentioned, a common approach in SDA is to employ contrastive learning, ignoring the geometry in the label space by embedding different input samples with the same label to a single embedded point. For classification, although ignoring the between-class geometry, such an approach may be adequate. In contrast to this approach, in our framework, the adaptation is performed by aligning the input-label joint distributions of the source domain to the target domain. This alignment is carried out by factorizing the input-label joint distribution to the marginal and conditional distributions. In the context of UDA, aligning the marginal distribution is a common practice. However, in the context SDA, we propose to directly align the conditional distribution (through the CDCA term), which is often overlooked in other SDA methods. As a consequence, our framework preserves the geometry of the input and label spaces, and hence, can be applied to a broad collection of learning tasks beyond classification, such as multi-label classification and regression. We have modified the "Related Work" section to convey these differences.
>
> 3. Regarding the choice of the architecture: note that in the CityCam experiment, we use ResNet-50, and in the Office experiment we use a VGG-16. We follow the same experimental protocol employed by the considered baselines [3,4]. The choice of the backbone, its architecture, and whether it is pre-trained or not, highly affect the results (for example, see the bottom line in Table I in [5], where the authors demonstrate the effect of the chosen architecture). Therefore, to fairly compare DA techniques, it is crucial to perform the evaluation with the same backbone. We note that in the literature, there are works that do not follow this guideline. For example, in dSNE, for the ``Digits'' datasets, instead of using the same backbone as in [3,6], a close look at the code reveals that LeNet++ was used [7].

---

> ### Author Response · Authors · 2024-04-24
> **Respones to Reviewer DCB5 - part 2**
>
> References:
>
> [1] Ben-David, S., Blitzer, J., Crammer, K., Kulesza, A., Pereira, F.,  Vaughan, J. W. (2010). A theory of learning from different domains. Machine learning, 79, 151-175.
>
> [2] Shore, J., \& Johnson, R. (1981). Properties of cross-entropy minimization. IEEE Transactions on Information Theory, 27(4), 472-482.
>
> [3] Saeid Motiian, Marco Piccirilli, Donald A Adjeroh, and Gianfranco Doretto. Unified deep supervised domain adaptation and generalization. In Proceedings of the IEEE international conference on computer vision, pp. 5715–5725, 2017b.
>
> [4] Xiang Xu, Xiong Zhou, Ragav Venkatesan, Gurumurthy Swaminathan, and Orchid Majumder. d-sne: Domain adaptation using stochastic neighborhood embedding. In Proceedings of the IEEE/CVF Conference on Computer vision and Pattern Recognition, pp. 2497–2506, 2019.
>
> [5] Lukas Hedegaard, Omar Ali Sheikh-Omar, and Alexandros Iosifidis. Supervised domain adaptation: A graph embedding perspective and a rectified experimental protocol. IEEE Transactions on Image Processing, 30:8619–8631, 2021.
>
> [6] Saeid Motiian, Quinn Jones, Seyed Iranmanesh, and Gianfranco Doretto. Few-shot adversarial domain adaptation. Advances in neural information processing systems, 30, 2017a.
>
> [7] Yandong Wen, Kaipeng Zhang, Zhifeng Li, and Yu Qiao. A discriminative feature learning approach for deep face recognition. In European conference on computer vision, pp. 499–515. Springer, 2016.

---

### Review · Reviewer_jTBJ · 2024-04-07

**Summary Of Contributions:**

This paper proposes an SDA approach by aligning the marginal and conditional distributions based on Ben-David et al. (2010). Specifically, the conditional distributions are aligned with kernel regression that takes into account the geometric structure of the input and label spaces. The model is evaluated on both classification and regression tasks.

**Audience:**

Yes

**Claims And Evidence:**

No

**Requested Changes:**

Overall, while I acknowledge that the authors made some good efforts to provide a clear exposition of their ideas, I found the core concept very similar to Ben-David et al. (2010) but less rigorous and theoretically motivated. I listed several concerns as follows.


- I think it would be better to clarify the difference from the SSDA task.

- How does Theorem 4.1. differ from Theorem 1 in Ben-David et al. (2010) which proposed:

$\epsilon_T(h) \leq \epsilon_S(h) + d_1(S,T) + \min(E_{x\sim S}|f_S(x)-f_T(x)|, E_{x\sim T}|f_S(x)-f_T(x)|)$

- How did you get $M$ out of the integral in Eq.12?


- The derivation from Eq.3 to E.4 is not theoretically motivated, where the classification loss of labeled target data is simply added to the objective.


- The theoretical analysis is incomplete, e.g., the generalization error of Eq.2 is missing.


- Where is the definition of $\ell$ in Eq.5?


- In UDA, we are forced to align marginal distributions $Q^S, Q^T$ since there are no target labels, which may increase joint error [1]. In this SDA setting, we can directly align $P^S_y, P^T_y$ to resolve the covariate shift. Therefore, I do not see the motivation to break this conditional alignment into marginal alignment ($Q^S=Q^T$) and CDCA ($P^S_x=P^T_x$).

- In that sense, I would like to see a baseline sorely based on conditional distribution alignment such as [2] to justify your motivation.


- How does your kernel regression method compare to simply training classifiers $h_S,h_T$ for $S, T$ and approximating $\hat{Y}^S=h_T(S),\hat{Y}^T=h_S(T)$?


- I doubt NEM or d-SNE published in 2019 is still SOTA.


- In Fig.3(b), the feature space seems not well-clustered. Consequently, I wonder if $\|\phi(X_i)-\phi(X_j)\|$ is still discriminative since the difference would be marginal for high dimensional vectors.


[1]On Learning Invariant Representations for Domain Adaptation, ICML 2019

[2]Conditional Adversarial Domain Adaptation, NIPS 2018

**Strengths And Weaknesses:**

Strengths:

- The conditional distributions are aligned with kernel regression that considers the geometric structure of the input and label spaces.

- The model is verified on both classification and regression tasks.

Weaknesses:

- The proposed Theorem 4.1. seems identical to that of Ben-David et al. (2010).

- Despite a theoretical approach, I do not observe any attempt to address the potential problem in this specific SDA setting, where limited labeled target data can increase generalization error.

- Unlike SSDA, SDA does not allow unlabeled target data during training. However, if my understanding is correct, you still need unlabeled target data to verify your model. This problem setting seems impractical in dealing with real-world tasks where unlabeled data is always available.

---

> ### Author Response · Authors · 2024-04-24
> **Response to Reviewer jTBJ - Part 1**
>
> We wish to thank the reviewer for these comments that helped us improve the manuscript. We have modified the manuscript to address the comments raised. Below is a list of the point-by-point responses and all the modifications we have made.
>
> 1. Regarding the differences between SSDA and SDA: in SDA, the training setup includes a small corpus of labeled data from the target domain and a large corpus of labeled data from a related source domain. Indeed, the setup of SSDA is more general and supports cases in which unlabeled target data from the target domain is also available. As we mentioned in the conclusion section, in future work, we will extend the proposed approach for semi-supervised domain adaptation. This is a natural extension of the proposed approach that follows the decoupling between the input and labels distributions in the derived bound.  However, we disagree that the setup of SDA is impractical. SDA is a contemporary field with a large body of research, e.g., [1-5].  Emerging domains (e.g. early stages of a pandemic), rare events (e.g. natural disasters), costly data acquisition (e.g. remote sensing or certain types of environmental monitoring), and privacy issues (e.g. in healthcare or financial sectors, data privacy regulations might prevent the use of even unlabeled data) are just a few scenarios where even unlabeled target data is out of reach and calls for an SDA approach. Following your comment we included these scenarios and motivations in the Introduction (highlighted in red).
>
>    [1] Saeid Motiian, Quinn Jones, Seyed Iranmanesh, and Gianfranco Doretto. Few-shot adversarial domain adaptation. Advances in neural information processing systems, 30, 2017a.
>
>    [2] Saeid Motiian, Marco Piccirilli, Donald A Adjeroh, and Gianfranco Doretto. Unified deep supervised domain adaptation and generalization. In Proceedings of the IEEE international conference on computer vision, pp. 5715–5725, 2017b.
>
>    [3] Xiang Xu, Xiong Zhou, Ragav Venkatesan, Gurumurthy Swaminathan, and Orchid Majumder. d-sne: Domain adaptation using stochastic neighborhood embedding. In Proceedings of the IEEE/CVF Conference on Computer Vision and Pattern Recognition, pp. 2497–2506, 2019
>
>    [4] Zengmao Wang, Bo Du, and Yuhong Guo. Domain adaptation with neural embedding matching. IEEE transactions on neural networks and learning systems, 31(7):2387–2397, 2019.
>
>    [5] Takeshi Teshima, Issei Sato, and Masashi Sugiyama. Few-shot domain adaptation by causal mechanism transfer. In International Conference on Machine Learning, pp. 9458–9469. PMLR, 2020.
>
>    [6] Lukas Hedegaard Morsing, Omar Ali Sheikh-Omar, and Alexandros Iosifidis. Supervised domain adaptation using graph embedding. In 2020 25th International Conference on Pattern Recognition (ICPR), pp. 7841–7847. IEEE, 2021.
>
>    [7] Antoine de Mathelin, Guillaume Richard, François Deheeger, Mathilde Mougeot, and Nicolas Vayatis. Adversarial weighting for domain adaptation in regression. In 2021 IEEE 33rd International Conference on Tools with Artificial Intelligence (ICTAI), pp. 49–56. IEEE, 2021.
>
> 2. Regarding the theoretical contribution of Theorem 4.1: Our theoretical contribution is two-folded. First, we generalize the definition of a domain from  Ben-David et al. (2010). In  Ben-David et al. (2010), a domain is defined as an *input* distribution equipped with a deterministic labeling function. We define a domain as an input-label *joint* distribution. Second, we generalize the result from  Ben-David et al. (2010) by considering arbitrary supervised learning tasks instead of just a binary classification. To highlight these contributions we added this paragraph to the manuscript. However, we wish to emphasize that the main contribution of our approach is not the theoretical derivation but harnessing it for SDA. The bound from  Ben-David et al. (2010) serves as the foundation for the vast majority of UDA methods, but in the context of SDA this bound was overlooked. Employing this bound in the context of SDA is a major contribution as it places, to the best of our knowledge, the *first* corner-stone for theoretically grounded and systematic algorithms for SDA.
>
> 3. Regarding the constant $M$:  we assume that the expectation of the loss function, denoted by $M$, is bounded. For example, for the $0-1$ loss function, $M$ is bounded by 1. In our proposed loss term, $M$ is multiplied by $d(\widetilde{\mathcal{Q}}^{\text{S}},\widetilde{\mathcal{Q}}^{\text{T}})$ which is minimized during the training process. Following your comment, we added this clarification to the manuscript.

---

> > ### Author Response · Authors · 2024-04-24
> > **Response to Reviewer jTBJ - Part 2**
> >
> > 4. Regarding the incorporation of the empirical target error (in eq. (2) and eq. (4)):  thank you for pointing out this issue (this point was also raised by reviewer 8Dhw). Indeed, including a noisy estimate of the target error and the weights of the loss terms are related. More specifically, for any $0 \leq \alpha \leq 1$, we can express the bound from eq. (2) as follows:
> >  $ \epsilon_{\mathcal{P}^T} (h) = (1-\alpha)\epsilon_{\mathcal{P}^T} (h) +\alpha \epsilon_{\mathcal{P}^T} (h)  \\ \leq  (1-\alpha)\epsilon_{\mathcal{P}^T} (h) +\alpha \big( \epsilon_{\mathcal{P}^S} (h)      + M d(\mathcal{Q}^T,\mathcal{Q}^S) + \min\{
> >      \chi^S,\chi^T\} \big). $   Expressing the bound from eq. (2) in this kind of form motivates the derivation of eq. (4), where the selection of the hyper-parameter $\alpha$ reflects the prior belief for the capability to estimate $h$ from the available target samples.  When target data is highly available, $\alpha$ can be set close to 0, and when target data is highly limited, $\alpha$ should be set close to $1$. In the revised manuscript we clarified this point (highlighted in red in Section 5.1).
> >
> > 5. $\ell$ from eq. (5) denotes the considered loss function. It is defined in Definition 3.2. Following your comment we added an explicit reminder in the text.
> >
> > 6. Regarding the motivation for factorizing the joint distribution:
> > In the manuscript, we used $\\mathcal{Q}^v, v\\in {S,T}$ to denote the marginal distribution of domain $v$, and $\\mathcal{P}_{x}^v$   to denote the conditional distribution of the output space given $x\\in \\mathcal{X}$ in domain $v\\in \\{S, T\\}$. We did not use the notation mentioned in the comment: $P^v_y, v \\in \\{S, T\\}$. Since it appears in the context of the CDCA term, we assume it is a typo and it should have been the conditional distribution $\\mathcal{P}_x^v$.
> > Indeed, the UDA term (responsible for aligning $\\mathcal{Q}^S$ and  $\\mathcal{Q}^T$) can be discarded.
> > However, factorizing the input-label joint distribution to the marginal and conditional distributions has two advantages. First, it facilitates the ability to align distributions of lower dimensions, which is an easier task compared to aligning distributions of higher dimensionalities. Second, an underlying assumption in SDA and SSDA is that the conditional distribution of the source domain is similar but not necessarily identical, to the conditional distribution of the target domain. In other words, there exists a prior on the similarity between the conditional distributions (the task), assuming the domain shift between the domains is mainly due to the covariate shift. Discarding this prior might result in inferior performances.
> > For example, in [1] the authors propose to directly align the joint distribution of the labels and the inputs. We did not consider [1] in our experimental study as it exhibited inferior results compared to [2], which is part of the baselines we consider. In addition, we wish to remark that empirically,  discarding the UDA term results in inferior performance, as can be seen in Table 1 by "Ab-UDA".
> >
> >    [1] Koniusz, Piotr, Yusuf Tas, and Fatih Porikli. "Domain adaptation by mixture of alignments of second-or higher-order scatter tensors." Proceedings of the IEEE conference on computer vision and pattern recognition. 2017.
> >
> >    [2] Saeid Motiian, Marco Piccirilli, Donald A Adjeroh, and Gianfranco Doretto. Unified deep supervised domain adaptation and generalization. In Proceedings of the IEEE international conference on computer vision, pp. 5715–5725, 2017b.
> >
> > 7. Regarding the choice of kernel regression rather than a parametric estimator for each domain: $\hat{Y}_S = h_T (S)$  and $\hat{Y}_T = h_S (T )$. The CDCA term cannot be approximated directly based on the observed data.
> > Computing the CDCA requires access to pairs of input-label samples (x,y), where y is sampled from the cross-domain of x, while in practice, we have only access to pairs of samples where x and y are sampled from the same domain. We address this challenge by generating “pseudo-labels” using non-parametric kernel regression. The motivation behind this particular choice, rather than a standard parametric estimator like a neural network, is to avoid overfitting due to the small number of available training samples from the target domain.
> >
> > 8. Regarding the selected baselines: to the best of our knowledge, NEM and d-SNE are considered as SOTA.

---

> ### Author Response · Authors · 2024-04-24
> **Response to Reviewer jTBJ - Part 3**
>
> 9. Regarding the contribution of the UDA term as it appears from the embedding in Fig.3(b). First, we wish to clarify that in Fig. 3(b) we presented the t-SNE embedding of the *raw-datapoints* of a colored MNIST dataset and not the embeddings which are induced by $\phi(\cdot)$. The purpose of the colored digits experiment was to demonstrate our method's ability to preserve the dataset's intrinsic geometry. Observing Fig.3(b), we can see two dominant geometrical structures. The first geometrical structure is induced by the digits presented in each image. The second one, which is the most dominant, is induced by the color of the digits in each image. As noticed in the comment, the discriminability of the raw feature space is limited with respect to the digits in each image. However, with respect to the color of each image, it is highly discriminative, and the dataset is highly clustered according to the colors of the image. Contrastive methods, like CCSA, would ignore this geometry as they try to embed different input samples with the same label to a single embedded point. To quantify the ability of our method to preserve the intrinsic structure, in addition to measuring accuracy, we measure the proportion of partial errors, where the color of the predicted classes is different than the color of the true class. Observing the results in Table 6, we can see that our method managed to yield significantly lower values of out-of-color errors compared to the contrastive baseline, especially when considering datasets with a large number of colored classes.

---

### Review · Reviewer_8Dhw · 2024-04-15

**Summary Of Contributions:**

This manuscript proposes a new method for supervised domain adaptation.  Specifically, the authors extend theory that uses an extension of theory from Ben-David on unsupervised domain adaptation to work with supervised domain adaptation, and then design an algorithm to minimize the theoretical bound.  This algorithm is run on a number of empirical tasks and compared to several existing algorithms, showing on-par or improved performance

**Audience:**

Yes

**Broader Impact Concerns:**

None.

**Claims And Evidence:**

No

**Requested Changes:**

## Necessary:

Include details of validation procedure and discussion of its reasonableness.

Modify draft so that key necessary details are contained within the main paper.

Increase discussion of related works (e.g., multi-task, related theory) to more clearly delineate contributions

Discuss reasonableness of second-order statistic estimation.

Provide results/discussion on different levels of data, including when source data is more available.

## Minor:

Reference format:  Several references should be (name, year) not name (year).

**Strengths And Weaknesses:**

## Strengths:

This manuscript is quite easy to read, and the claims are largely explicit.

The algorithm is compared on a variety of tasks (not just classification), especially useful since regression tends to act differently than classification.

I appreciated the honest discussion in the appendix about reproducibility and model selection are nice.

## Weaknesses/areas to improve:

### Backgrounds/Methods:

In Section 4, we the authors claim that they generalize the theory of Ben-David et al. (2010), and this is also noted in the introduction as a contribution.  However, it is unclear exactly which part is novel.  For example, [1] considered a generic loss function as an extension of the Ben-David based on the prior extension of Li et al. (2018).  The contribution in this section needs to be clarified.

[1] Richard, Guillaume, et al. "Unsupervised multi-source domain adaptation for regression."  ECML PKDD 2020.

Additionally, I do not see authors discuss multi-task learning at all.  There is so much overlap of concept between multi-task learning and supervised domain adaptation that it really should be discussed.

Equation 4 is not well motivated.  In particular, (2) is a lower bound on the target error, whereas adding in an estimate of the target error will loosen the lower bound. I get the motivation for using a noisy estimate of the target error, but this should be more carefully considered and explained.  Additionally, the equation now is basically: target error <= noisy realization of target error + bound on target error.  It seems like the different loss terms should be weighted.  For example, if there is a fair amount of target data, then it would seem prudent to use primarily the target error estimate rather than the bound.  Please add a discussion of this equation and whether weighting the samples is appropriate.

### Empirical evaluations:

The optimization approach requires second order statistics to estimate the loss terms.  It is surprising to me that this can be reasonably estimated by such a small number of target samples.  I would appreciate a section that elaborates on the errors and feasibility of that optimization with limited target samples.

One of the things that concerns me is the very precise settings to compare the algorithms.  For example, in the digits dataset the source data and target data are both highly limited, which is a scenario repeated throughout the empirical evaluation.  It leaves me with a question on whether the results hold conclusions hold broadly.  In particular, most UDA papers have a pretty large source domain, and most real-world problems I consider have a fair amount of source data.  The authors need to clearly motivate limiting the source data, and discuss this extensively.  Additionally, the authors are considering highly data-limited target regimes.  As someone who has applied SDA in practice, many of our real world problems we have hundreds to thousands of labeled target data, so I am unable to evaluate from these results whether I should consider their algorithm  I would the authors to more clearly elaborate on what happens in different data regimes, and clearly describe the scenarios where their method is beneficial.

Addition, the authors need to include more experimental details in the main paper.  I could not evaluate many of the claims without first reading through the appendix.  Key details need to be in the flow of the text.

The reproducibility of results is an interesting argument made in the appendix. In particular, the “Hyperparameter inconsistency” point argues that the same hyperparameters need to be tuned for each experiment.  I disagree with this assertion in general but understand the reasonableness in local scenarios.  I would like the authors to elaborate on this claim.  Secondly, as they point out, the lack of a validation set is a problem for tuning.  This highlights that the authors need to describe the validation and performance estimation approach very clearly.  While the authors note that they adopt the strategy of Hedegaard et al. (2021), the key details need to be repeated here so that the manuscript is more self-contained.  E.g., what data was used as validation, test, etc?  How well can the algorithms do with generic default parameters?

---

> ### Author Response · Authors · 2024-04-25
> **Response to Reviewer 8Dhw - Part 1**
>
> We wish to thank the reviewer for these comments that helped us improve the manuscript. We have modified the manuscript to address the comments raised. Below is a list of the point-by-point responses and all the modifications we have made.
>
> 1. Regarding the generalization of the theorem from  Ben-David et al. (2010): we generalize the result from Ben-David et al. (2010) by considering arbitrary supervised learning tasks instead of binary classification, and by defining the domain as an input-label joint distribution rather than an input distribution equipped with a deterministic labeling function.  In the context of UDA, aligning the marginal distribution is a common practice. However, in the context SDA, the alignment of the conditional distribution is often overlooked. In contrast to other methods, we propose to directly minimize the CDCA term, which is derived from the bound, and explicitly take into account the conditional distributions.
>
> 2. Regarding the relation to [1]: as we mentioned in the previous comment, in the context of *unsupervised* domain adaptation aligning the marginal distribution is a common practice. This is the exact case in [1]. In [1], the authors present the following bound: $ \\epsilon_t(h,f_t) \\leq  \\epsilon_k(h,f_k) + \\eta_{\\mathcal{H}}(f_k,f_t) + \\textit{HDisc}_{\\mathcal{H},L}(p_t,p_k;h) $.
>
>     At first glance, this bound seems very similar to the one we propose, as it also factorizes the input-label joint distribution to the marginal and conditional distributions.  However, a closer look reveals that unlike the bound we propose, where the joint distribution ($p(x,y)$) is factorized to the marginal distribution of the input space ($p(x)$, which yields the UDA term) and the conditional distribution of labels space ($p(y|x)$, which yields the CDCA term), in [1] the authors propose to factorize the joint distribution to the marginal distribution of the *output* space ($p(y)$, which yields the term  $\\eta_{ \\mathcal{H}}(f_k,f_t) $) and the conditional distribution of *input* space  ($p(x|h)$, which yields $\\textit{HDisc}_{\\mathcal{H}, L}(p_t,p_k;h)$, and $p(h|f)$ which yields $\\epsilon_k(h,f_k)$).
>
>    In [1], where the authors address the problem of *unsupervised* domain adaptation, the authors propose a method to minimize the conditional term,  and discard the minimization of $\\eta_{\\mathcal{H}}(f_k,f_t)$:
>
>     "The second term $\\eta_{\\mathcal{H}}$ is the sum of the error made by the ideal hypothesis on both domains: it is small when the two labeling functions are close which is the general assumption of unsupervised domain adaptation. As it involves $f_t$ it cannot be controlled in unsupervised domain adaptation without access to labels in the target domain". In contrast, we address the problem of *supervised* domain adaptation and devise a method to minimize both the UDA term and the CDCA term. Following your comment, we have modified the manuscript and included this reference and remarks (highlighted in red in Section 1).
>
>    [1] Richard, Guillaume, et al. ”Unsupervised multi-source domain adaptation for regression.” ECML PKDD 2020.
>
>
> 3. Regarding the relation between supervised domain adaptation and multi-task learning: we acknowledge that these two fields share several similarities, including the goal of improving model performance through additional supervision and leveraging information from related tasks. We modified the manuscript and addressed the relationship between SDA and MTL (highlighted in red in the Related Work Section).
>
> 4. Regarding Equation 4 and the motivation for including the target error: thank you for pointing out this issue. Indeed, including a noisy estimate of the target error and the weights of the loss terms are related. More specifically, for any $0 \\leq \\alpha \\leq 1$, we can express the bound from eq. (2) as follows: $ \\epsilon_{\\mathcal{P}^T} (h) = (1-\\alpha)\\epsilon_{\\mathcal{P}^T} (h) +\\alpha \\epsilon_{\\mathcal{P}^T} (h)  \\leq  (1-\\alpha)\\epsilon_{\\mathcal{P}^T} (h) +\\alpha \\big( \\epsilon_{\\mathcal{P}^S} (h) + M d(\\mathcal{Q}^T,\\mathcal{Q}^S) + \\min\\{\\chi^S,\\chi^T\\}\\big)$.  Expressing the bound from eq. (2) in this kind of form motivates the derivation of eq. (4), where the selection of the hyper-parameter $\alpha$ reflects the prior belief for the capability to estimate $h$ from the available target samples. When target data is highly available, $\alpha$ can be set close to 0, and when target data is highly limited, $\alpha$ should be set close to $1$. In the revised manuscript we clarified this point (highlighted in red in Section 5.1).

---

> ### Author Response · Authors · 2024-04-25
> **Response to Reviewer 8Dhw - Part 2**
>
> 5. Regarding the estimation of the second order statistics: you are correct, the approximation of $\\hat{\\epsilon}_{\\widetilde{\\mathcal{P}}^T}$ from eq. (4) is limited due to the small number of labeled training datapoints from the target domain. However, empirically, as already noted by other methods as well [1,2], incorporating this (noisy) estimation in the loss term is beneficial. The same remark holds for the approximation of  $\\min\\{ \\widetilde{\\chi}^{S},\\widetilde{\\chi}^{T}\\}$. Our empirical results show that incorporating the CDCA term serves as a good regularization that helps the model to achieve good generalization on the target domain. We note that due to the small number of training samples from the target domain, parametric estimators for the CDCA term are prone to overfitting. Therefore, we suggest implementing the CDCA term using non-parametric kernel regression (KR). Note that the KR is applied to the vectors in the low-dimensional embedded space $\\mathcal{E}$.
>
>    [1] Xu, X., Zhou, X., Venkatesan, R., Swaminathan, G., and Majumder, O. (2019). d-sne: Domain 403 adaptation using stochastic neighborhood embedding. In Proceedings of the IEEE/CVF Conference 404 on Computer Vision and Pattern Recognition, pages 2497–2506.
>
>    [2]  L.Hedegaard, O.A.Sheikh-Omar, and A.Iosifidis,“Supervised domain adaptation using graph embedding,” International Conference on Pattern Recognition,2021.
>
> 6. Regarding the precise settings of the source data: in the experiments that appear in Section 6.1 (Classification on benchmark tasks), we followed the exact experimental protocol employed by the considered baselines (see Section 5.1.2 in [1] and Section A.2 in [2]). We acknowledge that in most of the real-world problems, the data from the source domain is considered to be unlimited, but for the sake of a fair comparison, we chose to adopt the common evaluation protocol. Note that for the VisDA-C and the CityCam experiments, the entire source dataset is used for the evaluation. In addition, in the experiments we present in Section  6.2 (Leveraging the intrinsic geometry of a dataset) we also use the entire source datasets for the evaluation.
>
>    [1] Saeid Motiian, Marco Piccirilli, Donald A Adjeroh, and Gianfranco Doretto. Unified deep supervised domain adaptation and generalization. In Proceedings of the IEEE international conference on computer vision,  pp. 5715–5725, 2017b.
>
>    [2] Zengmao Wang, Bo Du, and Yuhong Guo. Domain adaptation with neural embedding matching. IEEE transactions on neural networks and learning systems, 31(7):2387–2397, 2019.
>
> 7. Regarding the experimental details: for the sake of brevity we included the experimental details in the SM. We appreciate this suggestion and moved key details to the main paper accordingly (highlighted in red).

---

> ### Author Response · Authors · 2024-04-25
> **Response to Reviewer 8Dhw - Part 3**
>
> 8. Regarding hyperparameters selection: in Section B in the SM (Baselines  Reproducibility on Benchmarks Dataset) we discuss the challenges associated with hyperparameters selection in two places. First, when discussing dataset partitioning. In the context of supervised domain adaptation, the definition of a validation set is not straightforward and there is no standard practice for it. For example, in the extreme case of 1 sample per target class, it is unclear how to partition samples for validation. Inspecting the code of recently published papers shows that the prevalent practice is to use the test set as validation. This practice leads to biased evaluations and overestimating the model performance, as the model is exposed to the test set, undermining its ability to generalize to unseen data. Moreover, this practice also imposes two challenges for a fair evaluation. The first involves the potential risk of data leakage. This concern was recently raised by the authors of [1], who proposed a rectified experimental protocol and released a Python package for benchmark datasets. This protocol is yet to be considered as a standard, however, in our experiments, we adopted their protocol as we wish to encourage others to do so as well. Nevertheless, data leakage is not the only challenge imposed by considering a validation set in the context of SDA. The ability to utilize target samples from the validation set provides additional information regarding the target domain that may not be applicable in a real setup. This additional information is not negligible, especially when considering a setup with an extremely small number of samples from the target domain. Therefore, when employing hyperparameter optimization, it is important to ensure that the same computational budget is used in the hyperparameter tuning of each method. Otherwise, the ability to perform more "peeks" to the target domain via the validation set gives an unfair advantage to models that were tuned using extensive hyperparameter searches. This is the case in [1], where the authors employ an extensive and systematic hyperparameter search (using  Bayesian Optimisation) only to their method. Since we could not employ a systematic hyperparameter search for all the methods (due to limited computational resources), but could not rely on the reported results, we chose to face this difficulty by avoiding any systematic hyperparameter search. The validation set was used only to find a coarse working point, and we used the same hyperparameters for all the considered methods. We acknowledge that this procedure is not ideal, however, we did our best to provide an honest comparison.
> Concerning your question - for the evaluation of the benchmark datasets we used the Python package provided by [1], where the data for the validation is taken as the rest of the dataset that was not picked for testing.
>
>    [1]  Lukas Hedegaard, Omar Ali Sheikh-Omar, and Alexandros Iosifidis. Supervised domain adaptation: A graph embedding perspective and a rectified experimental protocol. IEEE Transactions on Image Processing, 30:8619–8631, 2021.

---

> > ### Comment · Reviewer_8Dhw · 2024-05-10
> > **Response to authors**
> >
> > Thank you for your detailed response, which was mostly satisfying.  I would suggest adding comments on your response points 5 & 6 into the manuscript, which I didn't see highlighted in the red text.  If I simply missed it, I would appreciate you pointing it out.
> >
> > Regarding point 6, I am confused by the response given versus the text.  You stated in the response 6, "In addition, in the experiments we present in Section 6.2 (Leveraging the intrinsic geometry of a dataset) we also use the entire source datasets for the evaluation."  In Section 6.2 of the draft, it states, "In each experiment, 200 samples per class from the source domain and one sample per class from the target domain are randomly selected as the training set."  Validations, as I understand it, is on the target domain.   I may be misreading these statements, but these don't seem to be saying the same things.  Can you please clarify?
> >
> > I do have a remaining question on the impact of number of samples in the source domain. As it is, I do not know how these algorithms perform as the number of data samples changes, and that is a point that I would personally find interesting.  That may be beyond the scope here, but I would encourage the authors to consider it.

---

> > > ### Author Response · Authors · 2024-05-12
> > > **Response to Reviewer 8Dhw**
> > >
> > > Thank you once again for your detailed feedback on our manuscript. We appreciate your time and effort in reviewing our work.
> > >
> > > 1. Regarding point #5, in the fourth paragraph of Section 5.2 (Implementation) we referred to Appendix D for a further discussion on the computational complexity and limitations that stem when using kernel regression. Following your request we moved this discussion to the main text (highlighted in red).
> > >
> > > 2. Regarding point \#6, indeed, in the first experiment in Section 6.2 we followed the common experimental protocol and used only 200 samples per class from the source domain. However, in the rest of the experiments in this section (the  Multilabel-MNIST, Colored-Digits, CityCam, and Gasoline dataset) we used the entire source datasets for the evaluation.
> > >
> > >    Following this comment, we added a remark to clarify this point (highlighted in red).
> > >
> > > 3. Based on our empirical tests, our algorithm, along with other algorithms for SDA, behaves similarly when considering more samples from the source domain, where the main bottleneck is the number of samples from the target domain. Therefore, despite the importance of this issue, we believe that a rigorous analysis of the impact of the number of samples in the source domain on the generalization ability in the target domain exceeds the scope of this paper as it is not specific to our algorithm and calls for a holistic treatment that involves the entire field of SDA.

---

### Decision · Action_Editor_2Q1v · 2024-06-11

**Recommendation:** Accept with minor revision

**Comment:**

The paper propose to address the problem of supervised domain adaptation (SDA) by minimizing an upper bound inspired by the work on Ben David et al. in unsupervised domain adaptation. The authors describe how to implement the proposed approach (using a kernel regression on the labels) and present experiments on classification, multi-label and regression DA to highlight the generality of the method.

The reviewers found the paper interesting and well written but had several major comments and questions. The authors clarified most of the questions and updated the paper accordingly. The answers were appreciated by all reviewers but a few remaining concerns were raised during the final discussion. First the method is quite incremental wrt the bound of Ben David, and the assumptions on L prevent the used of the most commonly used classification loss  and the bound does not seem to be very tight with missing comparison to existing results. Also the positioning of the SDA method wrt other existing work especially Unsupervised DA and Multi-task learning is still lacking despite some localized changes and more references in the paper. The last concern was the limited performance gain in the numerical experiments compared to competitors.

Despite those concerns that limit the potential impact of the paper, it is well written and the claims are reasonable and justified. The approach is also of interest to the community so the paper does respect the requirement for publication in TMLR. But some remaining work in positioning  is necessary before acceptance :
+ The positioning of the approach wrt existing method especially multi-task learning and multi-source DA should be done with more details that adding a few references. The SDA problem as described can be solved with MTL and MSDA so a description of what is done in the literature and why the proposed approach is different is necessary. For instance the proposed approach looks similar to [A] and [B] that minimize the other conditional distribution in the embedding.
+ As noted by a reviewer it is important to position and discuss more in detail the interest of the proposed SDA approach wrt the similar Margin Disparity Discrepancy from [C].

[A] Gong, Mingming, et al. "Domain adaptation with conditional transferable components." International conference on machine learning. PMLR, 2016.

[B] Heinze-Deml, Christina, and Nicolai Meinshausen. "Conditional variance penalties and domain shift robustness." Machine Learning 110.2 (2021): 303-348.

[C] Zhang, Yuchen, et al. "Bridging theory and algorithm for domain adaptation." International conference on machine learning. PMLR, 2019.

**Audience:**

The results and proposed approach  would be of interest to TMLR's audience

**Claims And Evidence:**

The claims in the paper (proposition of an extension of the results of Ben David and generality of the method in practical applications) are well supported by theoretical proof and numerical evidence. The performance gain is limited but presented as such in the paper.

---

> ### Author Response · Authors · 2024-07-08
> **Response to Action Editor 2Q1v**
>
> We would like to thank again the action editor and the reviewers for reviewing our manuscript. We want to express our appreciation for the time and effort dedicated throughout the revision process and for the insightful comments and constructive remarks that have greatly improved the quality of the manuscript.
>
> We were glad to read the positive feedback, and we have carefully addressed the remaining comments noted by the action editor and modified our manuscript accordingly.
>
> In the revised manuscript, we have incorporated a detailed positioning of our approach with respect to existing methods, focusing on multi-task learning and multi-source domain adaptation. Specifically, we now discuss the mentioned references [A] and [B] and their particular relation to our approach in terms of the consideration of the conditional distribution. In addition, we position and discuss in more detail the relation between our approach and [C]. [C] extends the theoretical result of Ben-David et al. (2010) to multi-class classification and introduces the Margin Disparity Discrepancy (MDD). Similar to our work, [C] is based on the generalization of the theoretical results of Ben-David et al. (2010). However, [C] addresses the challenge of UDA, and the proposed loss term, which is based on classification margins, is limited to classification tasks only.
>
> The modifications are in Section 1 (Introduction) and Section 2 (Related Work), and for convenience, they are highlighted in blue.